# Human brain integrates both unconditional and conditional timing statistics to guide expectation and behavior

Yiyuan Teresa Huang[1,2,3]*, Zenas C. Chao[1]*

1 International Research Center for Neurointelligence (WPI-IRCN), UTIAS, The University of Tokyo, Tokyo, Japan, 2 Department of Multidisciplinary Sciences, Graduate School of Arts and Sciences, The University of Tokyo, Tokyo, Japan, 3 Japan Society for the Promotion of Science (JSPS), Tokyo, Japan

* yyhuang.teresa@gmail.com (YTH); zenas.c.chao@gmail.com (ZCC)

## Abstract

Our brain uses prior experience to anticipate the timing of upcoming events. This dynamic process can be modeled using a hazard function derived from the probability distribution of event timings. However, the contexts of an event can lead to various probability distributions for the same event, and it remains unclear how the brain integrates these distributions into a coherent temporal prediction. In this study, we create a foreperiod sequence paradigm consisting of a sequence of paired trials, where in each trial, participants respond to a target signal after a specified time interval (i.e., foreperiod) following a warning cue. The prediction of the target onset in the second trial can be based on two probability distributions: the unconditional probability of the second foreperiod and its conditional probability given the foreperiod in the first trial. These probability distributions are then transformed into hazard functions to represent the unconditional and conditional temporal predictions. The behavioral model incorporating both predictions and their mutual modulation provides the best fit for reaction times to the target signal, indicating that both temporal statistics are integrated to make predictions. We further show that electroencephalographic source signals are also best reconstructed when integrating both predictions. Specifically, the unconditional and conditional predictions are encoded separately in the posterior and anterior brain regions, and integration of these two types of predictive processing requires a third region, particularly the right posterior cingulate area. Our study reveals brain networks that integrate multilevel temporal information, offering insight into the hierarchical predictive coding of time.

## Introduction

Predicting when an event will occur is a fundamental cognitive process that allows for the efficient allocation of attentional resources and optimal motor performance

**Data availability statement:** The raw data, along with the data underlying each table, figure, and supporting table and figure, are publicly available in the Open Science Framework (https://doi.org/10.17605/OSF.IO/VEDHP).

**Funding:** This study was supported by World Premier International Research Center Initiative (WPI), MEXT, Japan (to Z.C.C.) (https://www.jsps.go.jp/english/e-toplevel/) and Japan Society for the Promotion of Science, Japan (to Y.T.H) (https://www.jsps.go.jp/english/e-pd/). The funders had no role in study design, data collection and analysis, decision to publish, or preparation of the manuscript.

**Competing interests:** The authors have declared that no competing interests exist.

**Abbreviations:** AIC, Akaike's Information Criterion; BIC, Bayesian Information Criterion; EEG, electroencephalogram; HF, hazard function; mTRF, multivariate temporal response function analysis; TRF, temporal response function.

[1–3]. It has also been found that precise temporal predictions optimize the processing of sensory information by enhancing contrast sensitivity [4]. To study temporal prediction, researchers often use a foreperiod task where participants are required to press a button as soon as a target signal appears following a warning signal. The time interval between the warning and the target signals is referred to as the foreperiod. To achieve a fast response, it is crucial to learn the probability distribution of the foreperiod as participants can use it to expect the target onset. In this task, temporal prediction is often modeled by a hazard function (HF), which represents the ongoing updates to predictions as time progresses [5,6]. This function, derived from the probability distribution of the foreperiod, describes how the probability of the target signal occurring is updated over time, given that it has not yet occurred.

For neural correlates of the temporal prediction, higher hazard values (i.e., stronger prediction) have been associated with increased activity in the parietal area, as observed in single-neuron recordings and human MRI studies [7,8]. Additionally, increased alpha power in human electroencephalogram (EEG) has also been linked to higher hazard values [9]. Moreover, as temporal predictions elapse over time, these dynamics can be tracked in the brain by modeling time-resolved EEG signals in a forward encoding model. This modeling approach can effectively distinguish brain signals associated with varying HFs derived from different probability distributions [10]. However, the timing of an event may be influenced by multiple probability distributions, particularly when considering specific contexts involved. Take Beethoven's Symphony No. 5 as an example. It begins with a "short-short-short-long" motif, commonly known as "fate knocking at the door." To predict the fourth interval, one might expect a higher chance of "short" when only considering the probability distribution of each element, but there could be a higher chance of "long" when considering the conditional probability of the multi-element pattern that characterizes the motif. It raises a question of how the brain integrates various probability distributions for the same event across different levels, to generate a coherent temporal prediction.

To understand how the temporal prediction is established by probability distributions of event timings across multiple levels, we introduce a foreperiod sequence paradigm. In this paradigm, two foreperiod trials (FP1 and FP2) are paired in a sequence, leading to two levels of statistical regularities. Predicting FP2 can be based not only on the probability distribution of the single foreperiod, but also the probability distribution of FP2 conditioned on the preceding foreperiod (i.e., FP1) when the sequence structure is considered. We obtain two HFs from the two probability distributions to represent unconditional and conditional temporal predictions, respectively, and then model reaction times to the target and EEG source signals. It is worth noting that our paradigm aims to investigate multi-level temporal prediction, similar to the music example provided above. However, participants are required to actively respond to each target signal rather than passively listen. While this design may limit the applicability of our findings to real-world scenarios where an active response is not always required, it is essential for testing the integration of multiple statistics at the behavioral level.

Using a model-fitting approach, both behavioral and neural results indicate that the two statistics are learned jointly, rather than independently, for prediction establishment. Furthermore, we find that unconditional and conditional temporal predictions are encoded in distinct brain regions, while additional regions are necessary for processing their integration. Our study offers an experimental platform for exploring multilevel temporal prediction, and identifies key brain regions involved in the hierarchical predictive coding of time.

## Results

### Foreperiod sequence paradigm to establish multilevel temporal predictions

To control the predictability of an upcoming event based on two distinct probability distributions calculated at different levels, we created a novel foreperiod sequence paradigm. During each trial, participants received a warning signal (a low-pitched tone and a white dot on the screen) and were instructed to press the button promptly as the target signal (a high-pitched tone and a red dot) appeared. The delay between the warning and the target signal is referred to as the foreperiod (Fig 1A). Two consecutive trials are considered a "sequence" when the warning signal from the second trial occurs 1.2 s after the target signal from the first trial, with the interval between two consecutive sequences ranging from 3 to 3.2 s (Fig 1B, top). Within each sequence, the foreperiods for the first and the second trials are denoted as FP1 and FP2, respectively. This sequence design allows the predictability of FP2 to be established with two distinct probability distributions (Fig 1B, bottom). The first is an unconditional probability distribution, derived directly from the frequency of the single foreperiod (either FP1 or FP2), regardless of the sequence structure. The second is a conditional probability distribution, incorporating sequence structure by calculating the conditional probability of FP2 given FP1, the preceding trial.

Four sequence blocks were created, each with 50 sequences (100 trials) and a unique configuration of unconditional and conditional probability distributions (Fig 1C). The block design allows us to disentangle the effects of unconditional and conditional probability distributions. For example, in Block 1, the foreperiods were set in the range of 0.4–2 s, with a step of 0.1 s and exclusion of 1.2 s. To simply explain the design in the following, we categorized the foreperiods as short (between 0.4 and 1.1 s, denoted as S) or long (between 1.3 and 2 s, denoted as L). Two sequence types were used: LL (long FP1 and long FP2) and SS (short FP1 and short FP2), each presented 25 times. For the unconditional probability, there were 25 trials of S and 25 trials of L, resulting in an unconditional probability distribution of 50% for S and 50% for L (denoted as 50–50, Fig 1D, left column). For the conditional probability, long FP2 always followed long FP1, and thus the conditional probability distribution for FP2 given long FP1 was a skewed unimodal distribution of almost 100% L (denoted as 0–100, Fig 1D, middle column). Similarly, short FP2 always followed short FP1, and thus the conditional probability distribution for FP2 given short FP1 was a skewed unimodal distribution of almost 100% S (denoted as 100–0). The other three blocks consisted of different numbers of sequences LL and SS, with additional sequences LS and SL. While the unconditional probability was always controlled at an even 50–50 ratio, as in Block 1, the conditional probability varied. It is important to note that the conditional probability of FP2 is computed based on each FP1 duration, as shown in Fig 1D (right column), rather than based on the short or long labels. Therefore, our paradigm can accommodate any probability distribution and is not restricted to a bimodal structure.

Next, to model temporal dynamics of the unconditional and conditional predictions, unconditional and conditional probability distributions in the four blocks (Fig 1D) were transformed into HFs denoted as $HF_U$ and $HF_C$, respectively (Fig 1E; see the formula in Methods). Please note that predictability of FP1 is solely determined by its unconditional probability and the subsequent $HF_U$, which are the same across all blocks.

Thirty-one participants (16 males and 15 females; age: $23 \pm 2.9$ years old, mean ± standard deviation) were included in the study. During the experiment, the four blocks were delivered in a random order and repeated three times, each with a different random order, resulting in a total of 12 block representations (run). At the end of each run, participants were asked to identify which sequence types were more frequent to maintain their engagement with the experiment. The accuracy was $87 \pm 4\%$ (mean ± std, $n = 31$ participants, chance level = 50%). During the task, we also recorded reaction time to

PLOS Biology

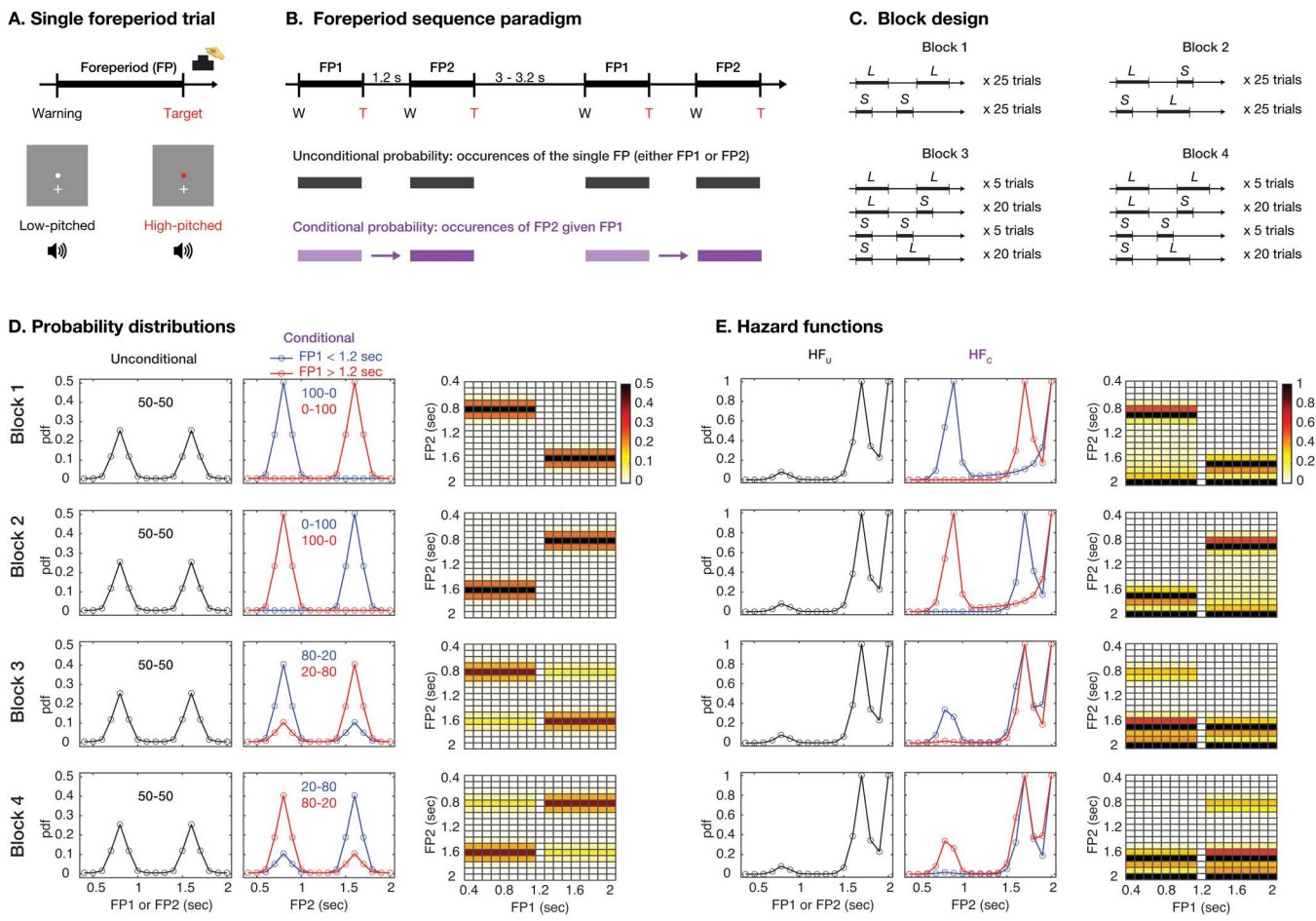

**Fig 1. Task design. (A)** A single foreperiod trial consisted of a white dot and a low-pitched tone as a warning signal, and a red dot and a high-pitched tone as a target signal. The foreperiod is the interval between the warning and the target, ranging from 0.4 to 2 s. Participants were instructed to press a *zero* button when the target appeared. **(B)** A sequence consisted of two foreperiod trials (denoted as FP1 and FP2) separated by a short blank, while a longer blank separated two sequences. There are two probability calculations. Regardless of the sequence structure, an unconditional probability can be computed based on single foreperiod occurrences (either FP1 or FP2) (highlighted with the black rectangle). A conditional probability can be computed based on FP2 occurrences (the purple rectangle) given on durations of FP1 (the light purple rectangle). **(C)** To vary the probabilities, four blocks were created, each containing 50 foreperiod sequences. The foreperiods timed between 0.4 and 1.1 s and between 1.3 and 2 s are denoted as *S* and *L*, respectively. There are four types of foreperiod sequences with different trial configurations for each of the four blocks. **(D)** Left column: The unconditional probability distributions of FP1 or FP2 are the same. Middle column: Two conditional probability distributions of FP2 were calculated. One was given FP1 < 1.2 s (in blue), and the other was given FP1 > 1.2 s (in red). The number on each plot (e.g., 50–50) represents the proportion of cumulative probabilities in the range of foreperiod > 1.2 s and foreperiod < 1.2 s. Right column: The unconditional and conditional probability distributions of each foreperiod were computed. The color indicates the probability value shown on the color bar. **(E)** The hazard functions (HFs) were computed based on the respective probability distributions in the panel C. $HF_U$ and $HF_C$ are computed based on the unconditional and conditional probability distribution, respectively. The data underlying this figure can be found in https://doi.org/10.17605/OSF.IO/VEDHP.

the target signal and 64-channel EEG signals. The data underlying each table and figure are available in a public repository on the Open Science Framework (https://doi.org/10.17605/OSF.IO/VEDHP).

## Reaction time modeled by both unconditional and conditional predictions and their interaction term

To investigate how unconditional and conditional temporal information can be used to predicting FP2, we analyzed the correlations between the values of $HF_U$ and $HF_C$ and the reaction times to the target following FP2 (later we simply refer to

as reaction times following FP2). The trials in which participants made responses before the target onset are classified as false alarm trials, with their frequency detailed in S1 Table. After excluding only the false alarm trials, the average reaction times following both FP1 and FP2 are shown in Fig 2 and S2 Table.

Before focusing on FP2, we first verified whether the reaction times following FP1 could be correlated with $HF_U$, as studies with a single foreperiod have shown that faster reaction times are associated with higher hazard rates [6,8,10]. To achieve this, we used a linear-mixed effect model to regress the reaction times following FP1 against the fixed effect, $HF_U$ values, and the individual difference was considered by including the participant as a random effect. A significantly negative effect of $HF_U$ was found, indicating that faster reaction following FP1 was associated with stronger prediction ($p < 0.001$, $n = 31$ participants, see Table 1). This is consistent with results from the literature. We also visualized the observed and predicted reaction times across all foreperiods (Fig 2A). A general trend consistent with the HF can be observed, particularly in the foreperiods with dominant trial frequencies, where actual and predicted reaction times closely align. Note that data from all blocks were combined for the model analysis; however, we separated them in the figure for clearer illustration of the actual reaction times across foreperiods alongside the predicted values.

Next, we examined how $HF_U$ and $HF_C$ contribute to the reaction time following FP2. We used linear-mixed effect models to regress the reaction times following FP2 against four different sets of variables: (1) $HF_U$ (denoted as $HF_U$-only), (2) $HF_C$ (denoted as $HF_C$-only), (3) $HF_U$ and $HF_C$ (denoted as $HF_U + HF_C$), and (4) $HF_U$ and $HF_C$ and their interaction term (denoted

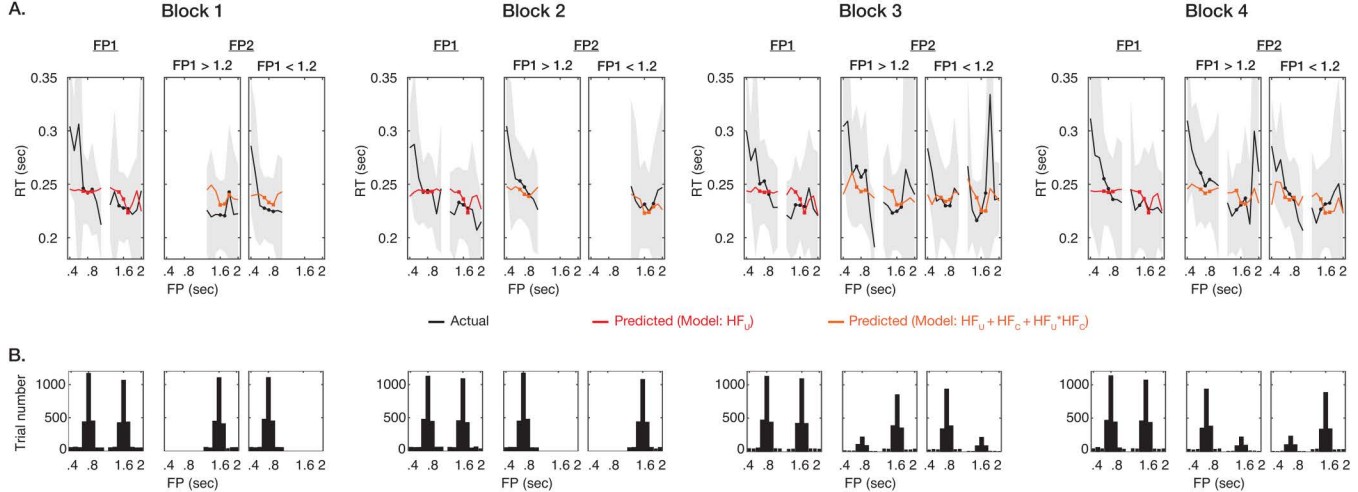

**Fig 2. Average and predicted reaction times across participants and their corresponding trial numbers. (A)** The average reaction time following FP1 and FP2 for each block is shown by the black line, with dominant foreperiod durations marked by black circles. The gray shade represents the standard deviation. Predicted reaction times are shown by the red line, with dominant foreperiod durations marked by red squares. For FP1, the predicted reaction times were obtained from a model with $HF_U$ as the fixed effect. For FP2, the predicted reaction times were based on a model including $HF_U$, $HF_C$, and their interaction as fixed effects. **(B)** The actual trial distributions after the removal of the false alarm are shown for FP1 and FP2 and blocks. The data underlying this figure can be found in https://doi.org/10.17605/OSF.IO/VEDHP.

**Table 1. Effect of $HF_U$ on reaction times following FP1.**

|  | Estimates | SE | β | *t* value | *p* | Con *R²* |
|---|---|---|---|---|---|---|
| (Intercept) | 0.244 | 0.006 |  | 42.88 | <0.001 | 0.173 |
| $HF_U$ | −0.021 | 0.002 | −0.08 | −11.67 | <0.001 |  |

*n* = 17,705 observations. Random effect: participants. Estimate: correlation coefficient. *β*: standardized beta coefficients. SE: standard error. Con *R²*: conditional *R*-squared.

as $HF_U + HF_C + HF_U * HF_C$). Please note that in our regression analysis, the duration of FP1 (*L* or *S*) was included as a random effect. This adjustment was necessary because we observed that the duration of FP1 influenced the response following FP2, which is known as the asymmetrical sequential effect [11]. Specifically, reaction times following a short FP2 were consistently longer after a long FP1 (as in the sequence *LS*) compared to after a short FP1 (as in the sequence *SS*) (see S3 Table). By including the duration of FP1 in the random effect, we were able to control for this confounding effect. Additionally, the participant was assigned as a random effect for controlling individual differences.

We first visualized the actual and predicted reaction times for all models in Figs 2 and S1. Due to the varied block-wise probability distributions, visually inspecting model fit is more challenging, as it requires jointly evaluating all predicted curves across the four blocks. To statistically determine the best-fitting model, we performed model comparisons. We find the model that incorporated $HF_U$, $HF_C$, and their interaction term (i.e., $HF_U * HF_C$) best predicts the reaction time following FP2 (with the smallest AIC and BIC for model fitness and the highest conditional *R*-squared for model explanation, see Table 2). The results of the best model are shown in Table 3, and the predicted reaction times are shown in Fig 2 (the predicted reaction times from the rest models are shown in S1 Fig). The interaction term had a notably positive coefficient ($p < 0.001$), indicating that the effect of $HF_U$ on the reaction time is moderated by the value of $HF_C$. Specifically, the effect of $HF_U$ on the reaction time was weaker under greater $HF_C$ values, but stronger under smaller $HF_C$ values. In other words, unconditional prediction had a reduced impact on behavioral responses when conditional prediction was strong. For $HF_U$ and $HF_C$, their main effects on FP2 were significantly negative ($p < 0.001$), indicating that higher hazard values, either unconditional or conditional predictions, led to faster responses. These findings suggest that the prediction of FP2 involves both $HF_U$ and $HF_C$. Importantly, unconditional and conditional predictions interact to influence behavioral responses, suggesting an integrative process that enables mutual modulation between the two predictions.

## Mapping unconditional and conditional temporal predictions to cortical activations

Building on the behavioral analyses, we further estimated how brain signals encode the unconditional prediction ($HF_U$) and the conditional prediction ($HF_C$) for FP2. Our approach was to identify brain areas where trial-by-trial EEG signals could be

**Table 2. Comparisons of linear mixed-effect models on reaction times following FP2.**

| Fixed effects | AIC | BIC | p | Con $R^2$ |
|---|---|---|---|---|
| ~$HF_U$ | −48388.5 | −48349.6 | <0.001 | 0.218 |
| ~$HF_C$ | −48335.4 | −48296.5 | <0.001 | 0.215 |
| ~$HF_U + HF_C$ | −48386.8 | −48340.1 | <0.001 | 0.218 |
| ~$HF_U + HF_C + HF_U * HF_C$ | −48449.7 | −48395.1 | | 0.221 |

*n* = 17,793 observations. Random effects: participants and FP1 durations. AIC: Akaike's Information Criterion. BIC: Bayesian Information Criterion. *p* value obtained by comparing the model in the current row to the one in the last row.

**Table 3. Effects of $HF_U$ and $HF_C$ on reaction times following FP2.**

| | Estimates | SE | β | *t* value | p | Con $R^2$ |
|---|---|---|---|---|---|---|
| (Intercept) | 0.243 | 0.007 | | 34.93 | <0.001 | 0. 221 |
| $HF_U$ | −0.049 | 0.005 | −0.21 | −10.44 | <0.001 | |
| $HF_C$ | −0.008 | 0.002 | −0.04 | −3.93 | <0.001 | |
| $HF_U * HF_C$ | 0.042 | 0.005 | 0.19 | 8.06 | <0.001 | |

*n* = 17,793 observations.

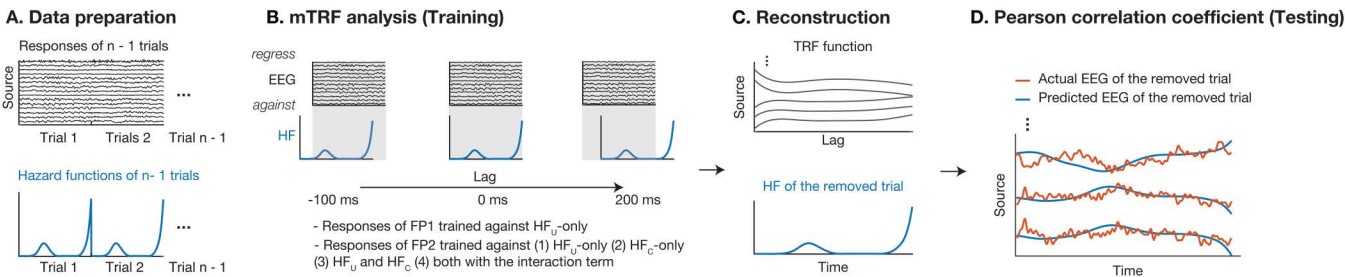

**Fig 3. Procedure for the forward-encoding model.** See the main text for details. The data underlying this figure can be found in https://doi.org/10.17605/OSF.IO/VEDHP.

reconstructed based on the HF over time. To achieve this, we used a forward encoding model, also known as regularized linear regression [12].

In our EEG analysis, to ensure sufficient data length for effective model training, we included trials that lasted > 0.7 s and did not contain a false alarm. These signals were then segmented, starting from 0.4 s after the onset of the warning signal and extending to the onset of the target signal. This segmentation minimized the influence of evoked responses to the warning and target signals. Then, we transformed EEG signals from the channel level (64 EEG channels) to the source level (~3,000 sources), using individual 3D electrode locations and structural MR images (see details of the source analysis in Methods). For $HF_U$ and $HF_C$, values from 0.4 to 2 s with a 0.1-s step (10 Hz) were splined-interpolated to share the same sampling rate as the EEG source signals at a rate of 250 Hz (see S2 Fig).

To identify which brain areas encode $HF_U$, $HF_C$, and their interaction term, we trained a forward-encoding model to reconstruct the EEG source signals from the hazard values. First, we modeled the EEG signals during FP1 on $HF_U$ only, which is similar to a typical single foreperiod scenario and was used as a benchmark. Then, we modeled the EEG signals during FP2 on (1) $HF_U$-only, (2) $HF_C$-only, (3) $HF_U + HF_C$, and (4) $HF_U + HF_C + HF_U*HF_C$, following the same approach used in the behavioral analysis.

The training was done for each trial and participant with a leave-one-out cross-validation approach. First, one trial was excluded and used as a testing trial, and the remaining trials were used for training (Fig 3A). For EEG signals, the training data comprised dimensions of $n − 1$ trials, $m$ time points, and ~3,000 sources. For the HFs, the training data comprised dimensions of $n − 1$ trials by $m$ time points. Then, we modeled the EEG signals from each source on the hazard values with different time lags (76 lags, from −100 to −200 ms with a 4-ms step), resulting in the temporal response function (TRF) (dimensions: ~3,000 sources and 76 lags) (Fig 3C). Each TRF value represents the weight of the HF on the EEG signal at a specific source and lag. Then, the source data were projected to the voxel level of the brain (181 * 217 * 181 = 7,109,137 voxels).

To evaluate the modeling performance, we used the TRF and the HFs from the testing trial (Fig 3C, bottom) to reconstruct the EEG signals for each source at a time lag of zero, focusing on the immediate effect of the HF on the EEG signals. We then calculated the Pearson correlation coefficients between the reconstructed and actual responses (Fig 3D) and averaged the correlation coefficients across all leave-one-out training (i.e., $n$ times from Fig 3A–3D) for each source for each participant. Sources with significant correlations across participants were identified as "significant areas" ($n = 31$ participants, Monte Carlo method, 1,000 permutations, cluster-based correction, two-tail, $α = 0.05$; see more details in Methods).

For FP1, the significant areas included the middle and posterior cingulate areas, superior and middle temporal areas, supramarginal area, calcarine, lingual, and fusiform (Fig 4A). In these significant areas, the TRF values were generally negative (see S3 Fig), indicating higher $HF_U$ values (stronger predictions) led to smaller responses. Note that our

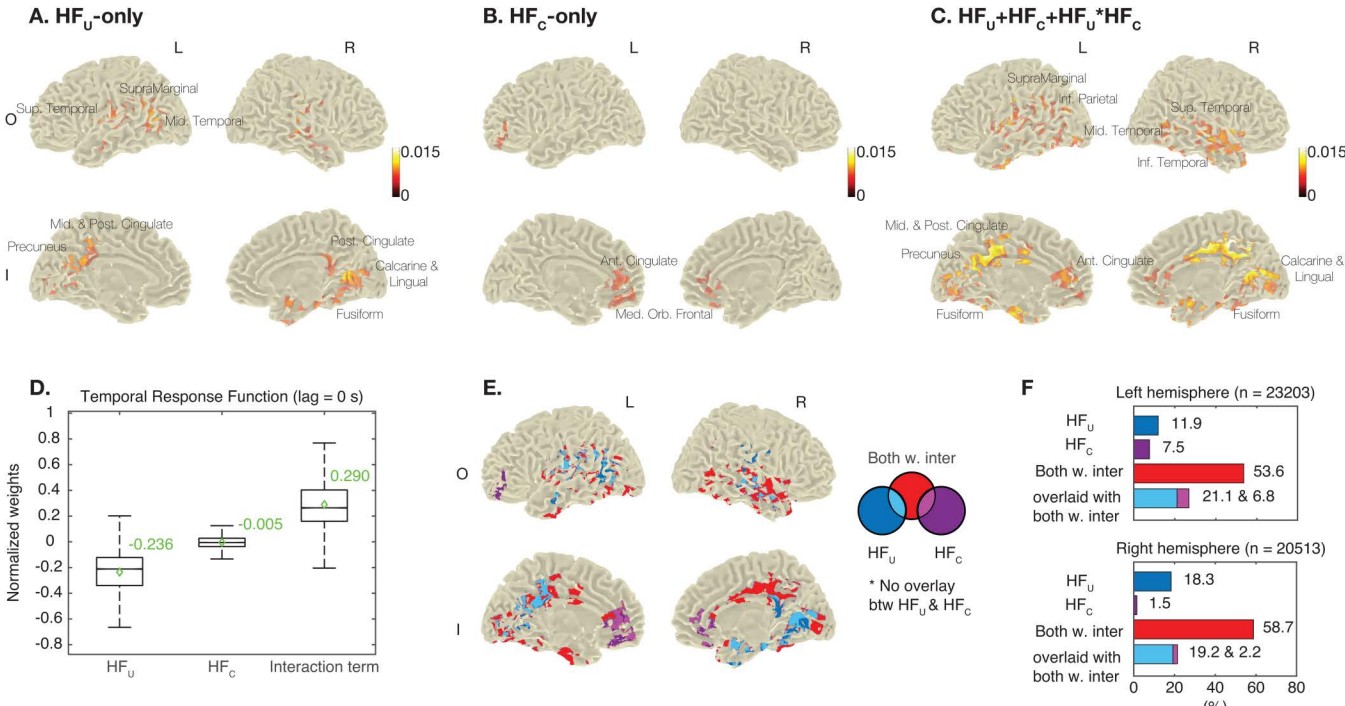

**Fig 4. Neural correlates for unconditional and conditional temporal predictions. (A)** EEG source responses during FP1 were modeled against $HF_U$-only. Four cortical surfaces with significant correlation coefficients are shown. The color bar shows correlation coefficients, and the same scale is applied to panels B and C. **(B)** EEG source responses during FP2 were modeled against $HF_C$-only, and cortical surfaces with significance are shown. **(C)** EEG source responses during FP2 were modeled against $HF_U + HF_C + HF_U * HF_C$, and cortical surfaces with significances are shown. **(D)** Within the significant areas in the panel C, the box plot shows the average TRFs between source responses and hazard values at a 0-s lag. Values were normalized between −1 and 1. In each box plot, there are the median (the middle horizontal line), quartiles (the bottom and top edges), and the maximum and minimum (the horizontal lines outside the box). **(E)** The significant areas in the panels A–C were compiled. There is no overlap between the significant areas for $HF_U$-only and $HF_C$-only. **(F)** The bar plots show the percentages for significant areas for each hazard function over all areas in the left and right hemispheres, respectively. The percentage is calculated using the sum of significant voxels in the panel E. The data underlying this figure can be found in https://doi.org/10.17605/OSF.IO/VEDHP.

prediction performances, indicated by the Pearson correlation coefficients between the actual and predicted signals, are below 0.015. While these may seem low, they fall within the range. For instance, the study by Di Liberto and colleagues [13] reported correlation values around 0.06 using EEG electrode signals in relation to speech characteristics, while Herbst and colleagues [10] found a correlation of approximately 0.005 when using EEG source data and HFs.

For FP2, significant areas were found only when modeling with (1) $HF_C$-only, and (2) $HF_U + HF_C + HF_U * HF_C$. The significant areas for $HF_C$-only included the anterior cingulate and medial orbitofrontal areas (Fig 4B). In these areas, we also observed negative TRF values (S4 Fig), indicating higher $HF_C$ values were associated with smaller responses. On the other hand, the significant areas for $HF_U + HF_C + HF_U * HF_C$ included the anterior, middle, and posterior cingulate areas, superior and middle temporal areas, supramarginal area, inferior parietal area, calcarine, lingual, and fusiform (Fig 4C). This training yielded significantly higher correlation coefficients compared to $HF_C$-only ($n = 31$ participants, Monte Carlo method, 1,000 permutations, cluster-based correction, two-tail, $\alpha = 0.05$). This indicates that the EEG signals were better reconstructed when incorporating $HF_U$, $HF_C$, and their interaction term, rather than considering $HF_C$ alone, which is consistent with the behavioral model comparisons (Table 2).

For these significant areas for $HF_U + HF_C + HF_U * HF_C$, we show the boxplots of the average TRF values for $HF_U$, $HF_C$, and their interaction term ($n = 35,104$ significant voxels for each TRF) (Fig 4D, also see S5 Fig for spatial distribution of

these values). First, the median TRF values for $HF_U$ and $HF_C$ were negative; again these indicate high unconditional and conditional predictions are associated with reduced responses. However, the median TRF value for their interaction term was positive, suggesting that the influence of $HF_U$ on the responses was weaker when the influence of $HF_C$ was strong, and vice versa. Again, these findings align with the beta values of the behavioral model.

We further compared the significant areas for $HF_U$-only (as shown in Fig 4A), $HF_C$-only (Fig 4B), and for $HF_U + HF_C + HF_U * HF_C$ (Fig 4C). Here, we refer to these as the $HF_U$ area, $HF_C$ area, and the integration area, respectively. In Fig 4E, we find that the $HF_U$ area and $HF_C$ area (colored separately in dark blue and dark purple) were mutually exclusive. Furthermore, the $HF_U$ area and the integration area were broadly overlaid (light blue, the intersection of dark blue and red), while the $HF_C$ area and the integration area were shared focally in the prefrontal area (light purple, the intersection of dark purple and red). For the integration area that was not shared with the $HF_U$ and $HF_C$ areas (colored in red), several regions were identified, including the middle and posterior cingulate areas, the inferior parietal area, and calcarine.

Fig 4F further illustrates the percentages of these areas in the left and right hemispheres. The integration area (colored in red) was found to be dominant, accounting for more than half of the total significant areas in both hemispheres. In summary, our findings indicate that the unconditional and conditional predictions are encoded in distinct brain regions, while the integration of these predictions involves large areas as well as additional regions.

## Discussion

Our foreperiod sequence paradigm was designed to create two levels of event timing regularities for temporal prediction. For both behavioral and EEG data, we demonstrated that predictions are made using statistics from both levels. Importantly, these statistics not only influence responses but also interact with each other. In other words, they are integrated to generate predictions. We further showed that these processes occur in distinct but overlapping brain regions. Our study establishes an experimental and analytical platform to isolate latent dynamics in hierarchical temporal prediction, a crucial step towards unifying predictive coding of "what" and "when" information (i.e., content and timing).

### Prefrontal cortex involvement in processing information at a longer time scale

Our investigation used a forward encoding model to estimate how the brain encodes unconditional and conditional temporal predictions based on learning regularities of the single foreperiod and foreperiod sequence. The HF is a representation of a prediction updated over time, and this dynamic characteristic was kept and assigned as a time-resolved regressor during model training. By training the model with $HF_U$ and $HF_C$ values, we disentangled their corresponding neural correlates and extracted the correlates of their interaction term at the source-level areas.

Specifically, only when the sequence structure was considered, the neural correlates in the anterior cingulate and medial frontal areas were identified. This prefrontal involvement has been reported in previous neuroimaging studies showing differences when comparing responses in the variable foreperiod paradigm (i.e., a uniform distribution of the foreperiod) and in the fixed foreperiod paradigm [7,14,15]. Additionally, in rodents, inactivations of bilateral dorsomedial prefrontal cortex led to premature motor responses, indicating top-down inhibition on the motor cortex during preparation [16,17]. While these previous studies localized prefrontal activity by manipulating single foreperiod occurrences (akin to the regularity of the single foreperiod in our design), we observed similar activity only during processing the foreperiod sequence. In fact, increased responses were found in ERPs during both FP1 and FP2, and a significantly positive waveform was found at the frontal electrodes when comparing ERPs during FP1 and FP2 (S6 Fig). The activation around these frontal electrodes may imply a function of sustained monitoring. For instance, in the previous studies, increased prefrontal activity was found in the condition of unimodal probability distributions, compared to the condition of precisely invariant FP. Also, the retention of temporal information over a longer scale is supported by prefrontal neurons activating differentially to store different durations of the first and second cues [18]. For our findings, ramping-up prefrontal activity may reflect tracking of FP1 and FP2 in the two-foreperiod sequence.

## A distinct brain network for the integration of unconditional and conditional temporal predictions

We used the multiplication of the $HF_U$ and $HF_C$ values to represent the integration of the unconditional and conditional predictions. This approach was inspired by the interaction term in regression models, which allows for the combined effect of two or more dependent variables on the independent variable, in addition to their individual main effects. Interestingly, in both behavioral regression and neural encoding models, we observed a positive effect of the interaction term. This indicates that the prediction at one level influences the reaction time and brain response more strongly when the prediction at the other level is weak, and vice versa. Furthermore, a wide range of brain areas was found to process integration, as there were no shared areas between significant correlates to $HF_U$ and $HF_C$. Particularly, the posterior cingulate cortex was notably identified with a large distribution among the important areas exclusively involved in the integration processing. The posterior cingulate cortex was part of the frontoparietal network comprising the precuneus and inferior frontal gyrus, and functional connectivity of this network mainly increased with update values in an ideal Bayesian observer model. The update is quantified as a divergence between prior and posterior probabilities of the current FP, compared to surprise, inversely and non-linearly correlated with the HF [19,20]. Furthermore, the neural correlate of the integration area localized preferentially at the right side of posterior cingulate cortex further strengthens a link to the top-down updates. Hemispheric lateralization in temporal processes has been proposed, where the right hemisphere would be better at learning previous information to predict future onset while the left hemisphere would be better at comparisons between a test stimulus interval and a target stimulus interval [21]. In sum, our finding of the cingulate area may imply simultaneously processing multiple levels of prior information, requiring intense prediction updates in the posterior cingulate cortex.

In addition to the cingulate areas, the inferior parietal area was also identified although its distribution was relatively sparse in our results. It has been shown that neural activity in the lateral intraparietal neuron field changed differently in blocks of unimodal and bimodal distributions [8]. Analogously, we created different bimodal distributions, e.g., the 50–50 unconditional probability distribution and 80–20 conditional probability distribution, and discrimination between the probability distributions may also be a function of the integration hub. Here, it is important to clarify again that the integration of two probabilities was supported by the model including the interaction term (i.e., $HF_U * HF_C$), which yielded the best fit relative to the other models that did not feature such a term. To directly investigate this integration effect within the brain, future research could alter activity in the identified areas through lesions, neuropharmacological manipulations, or brain stimulation techniques. By selectively enhancing or inhibiting each brain region associated with the $HF_U$ area, $HF_C$ area, or interaction area, we can assess its causal influence on behavior and on neural activity in the other regions.

Moreover, while our study provided insights into brain networks involved in temporal predictions based on time-series responses, it is also critical to understand how these networks function and communicate within the oscillatory domain. The alpha/beta oscillation has been suggested as a representation of the top-down prediction signal, while the gamma oscillation is associated with the bottom-up prediction error signal [22]. This aligns with findings that show that alpha power prior to a target changed with the HF [9]. Furthermore, predictions of "what will happen" at levels of single tone transitions and multi-tone sequences have been shown in different ranges of the beta oscillation [23]. To further investigate this frequency ordering of hierarchical predictions of "when an event will happen," we should combine time-frequency analysis as well as functional connectivity analysis in our future work. Moreover, while we transformed EEG surface signals into source signals to enhance spatial resolution, a better understanding of the areal brain functions could be achieved by combining EEG with functional magnetic resonance imaging measurements. Such a combination allows us to track the dynamics of temporal prediction signals and observe brain activity in precise locations.

We also acknowledge that our requirement of asking participants to identify the sequence structure may make the participants examine time more explicitly, particularly for learning the conditional probability, although the instruction was aimed to keep their engagement with the experiment. To explore implicit temporal predictions more effectively, the sequence identification question after each block can be removed. We anticipate that this modification may lead to the

extraction of integrating the two predictions in different brain areas, as well as different degrees of how the predictions modulate each other.

**Other potential temporal prediction models**

We noted that Janssen and Shadlen [8] adopted a smooth version of the HF, called temporal-blurred HF, assuming the precision of timing decreases as time goes on. On the other hand, recent studies using a probabilistically blurred probability distribution instead of a HF provided a better explanation for the reaction time [24,25]. Still, in long-foreperiod paradigms, where greater temporal uncertainty would be expected, reaction times have been well explained by temporally blurred HFs [26]. These seemingly conflicting findings across studies may reflect key differences across studies [27], including: (1) the statistical properties of the foreperiod distributions (e.g., mean and standard deviation), (2) the temporal resolution of the experimental design, and (3) the use of catch trials. These factors can introduce varied temporal prediction profiles with different types and magnitudes of uncertainty. Regarding the first point, in a bimodal distribution created by overlapping two identical unimodal distributions, a greater distance between the peaks is expected to produce a stronger temporal blurring effect. For probabilistic blurring, the extent of the effect can be influenced by the distribution's standard deviation, for instance, a sharper distribution may lead to more rapid changes over time than a flatter one. For the second point, even when distributions share the same mean and standard deviation, the range of foreperiods can still affect uncertainty. For example, a longer range (e.g., 0–20 s, as in [26]) introduces greater temporal uncertainty than a shorter range (e.g., 0.4–2 s, as used in our study). It is also reasonable to infer that temporal and probabilistic blurring may coexist and interact, potentially modulating one another, when using probability distributions with different means, standard deviations, and time ranges. Third, the inclusion of catch trials not only adds uncertainty regarding whether the target will occur but also complicates the definition of the underlying probability distribution. For instance, Grabenhorst and colleagues [25] modeled reaction times by incorporating both the uncertainty of target occurrence and probabilistic uncertainty (i.e., probabilistic blurring), raising the question of how these two forms of uncertainty interact in the brain. In terms of probability distributions, catch trials may either be excluded or interpreted as extremely long foreperiods, making distribution estimation more complex.

In our study, various smooth versions of HFs and probability distribution functions with different blurring parameters to fit the behavioral and neural data (see details in S1 Text) were tested. However, the results were not consistent: statistically significant improvement was observed for the reaction time (S7 Fig), but not the EEG data. Specifically, model performance quantified by the $R^2$ value varied depending on the type of blurring (temporally or probabilistically HF) and the dataset (FP1 or FP2), but the best models were obtained with HF as base. This inconsistency further supports our earlier points: in our study, we implemented distinct foreperiod distributions to examine whether the brain can learn and integrate both unconditional and conditional temporal statistics. This design inevitably introduced different types of uncertainty across conditions. As shown in previous neurophysiological work, which demonstrates neural encoding of original HFs, and consistent with our findings, we suggest that the brain may rely on HF-like computations as a neural mechanism for temporal prediction. These computations may be modulated by different forms of uncertainty in a context-dependent and potentially parallel manner. Nevertheless, we emphasize that while using nonblurred models allows us to isolate core computational principles, under the simplifying assumption that time is encoded without error, it also underrepresents the role of uncertainty. The modulatory effects of uncertainty should be explored in future research, for example, by varying foreperiod ranges (to examine temporal blurring) or manipulating the standard deviation of the distribution (to examine probabilistic blurring).

We also acknowledge that the conditional $R^2$ values in our behavioral data appear modest, and the relatively poor model fits can be attributed to several factors. First, we included reaction time outliers and used non-log-transformed data in our analysis (see details in Methods) to avoid introducing additional assumptions. We verified the contribution of trial-level noise from outliers to the model fit by demonstrating consistently improved conditional $R^2$ values when outliers

were removed (S3 Table). Second, unlike previous studies that used exponential or unimodal probability distributions, typically resulting in a single peak or a monotonic increase, we employed bimodal distributions that generate dual expectancy peaks. This probability structure produces more complex temporal expectations that cannot be adequately captured by a simple monotonic drift in brain signals. Third, to ensure a complete probability distribution structure, we included trials across the full foreperiod range in both our design and analysis, even though some foreperiod durations were in few trial numbers, likely introducing trial noise. This allowed us to more comprehensively capture the evolution of the HF over time. Fourth, our design is the first to create two probability regularities, and to achieve it, we also designed various bimodal distributions across blocks (e.g., equal 50–50 peak contributions versus asymmetric 20–80 peaks). Such complexity might increase trial-by-trial (also inter-individual) variability in learning and strategy, which in turn can increase behavioral noise and reduce the explained variance in models.

Finally, all the models we tested were framed within the HF approach and its derivative, the probability distribution, both of which rely solely on stimulus properties and lack a mechanistic foundation for how the brain processes them. More recently, several mechanistic computational and neural network-based models have emerged as promising alternatives, and their model predictions may better account for our observed data. For example, Bayesian frameworks describe temporal expectation as inference over time, with priors about event probability distributions continuously updated in light of new observations [20,28]. Similarly, reservoir computing models can be linearly or nonlinearly decoded to represent hazard-like signals or probabilistic expectations, making them strong candidates for biologically plausible temporal coding [29,30]. Future work may therefore benefit from incorporating these frameworks, which provide more mechanistic accounts of temporal expectation.

Still, we would like to emphasize that despite this added complexity, our key behavioral and neural findings remain robust. All major results, such as both significant behavioral and neural correlates with $HF_U$, $HF_C$, and the interaction term, were consistently verified and shown through control and additional analyses.

## The asymmetrical sequential effect observed in bimodal distributions

While our results demonstrated that reaction times were faster as hazard values were higher, we also observed an asymmetrical sequential effect due to our design composed of long/short FP1 paired with long/short FP2. The effect describes how reaction times increase if the preceding foreperiod trial is longer than the current one in the variable foreperiod paradigm. A model based on the principle of trace conditioning has been proposed to account for the effect [11]. Initially, in the case of a uniform probability distribution, the chance of a target occurring at any time is equal and thus the associated weights for each critical moment are equal. During the preparation for target occurrence, weights decrease as the corresponding critical moment passes (called extinction). This so-called extinction becomes weaker as time elapses, meaning that the weight associated with a late moment decreases less or remains relatively unchanged. At the imperative moment when the target is presented, the response is made, leading to an increase in the weight associated with the imperative moment (reinforcement). Then, the associated weights are passed to the next foreperiod trial. If the imperative moment is shorter (i.e., a target occurs after a shorter foreperiod than the previous one), the associated weight was already weaker for that moment, thus resulting in a slower response.

Regarding the asymmetrical sequential effect, two points should be noted: (1) Our regression results still show modulation from the HF while durations of FP1 were controlled as a covariate, and (2) Our forward encoding model results excluded the sequential effect. Regarding (1), we calculated the average reaction times following long and short FP2 after either short or long FP1 (i.e., *LL*, *LS*, *SS*, and *SL*) for four blocks (S8 Fig, the visualization of S4 Table). The asymmetry effect was observed, where reaction times following FP2 tended to be longer when FP1 had a longer duration (*LS* versus *SS*), but the effect was different among the blocks due to our manipulations for different probability distributions. For example, in S8 Fig, the asymmetry between the reaction times following FP2 in *LS* and *SS* is smaller in Block 4 but larger in Block 3. This is because of Block 4 composed of more trials of *LS* (40%) than Block 3 (10%), and fewer trials of

*SS* (10%) than Block 3 (40%). Therefore, to carefully identify the effect of the foreperiod on reaction time (i.e., temporal hazards) and exclude the sequential effect, we controlled the FP1 length in our regression analysis.

Regarding (2), it is known that anatomical locations responsible for the effect of the foreperiod on reaction and the sequential effect are distinct [31]. Patients with lesions to the right frontal area had a weaker foreperiod-reaction-time effect while the sequential effect remained intact. In contrast, lesions to the left premotor area diminished the sequential effect. Here, we adopted the forward encoding model and identified the neural correlates of HFs mostly in the posterior cingulate and frontal areas; however, an involvement of the premotor area was not detected. This suggests the forward encoding model as a plausible analysis tool to extract responses of temporal predictions while eliminating confounds due to the sequential effect. Still, although we used individual MRI images to reduce the volume conduction effect and reconstruct source-level signals, we should carefully consider the spatial limitations of the EEG when coming to a conclusion.

In conclusion, to our knowledge, this study is the first to reveal how the brain integrates multi-level information of event timing for prediction establishment. This also supports the hierarchical organization of the predictive-coding theory in the "when" domain. This paradigm was initially inspired by the auditory local-global oddball paradigm which was aimed to study the multi-level prediction of event pitches in the "what" domain [32,33]. As we solely manipulated the event timing (i.e., the "when" domain), in order to generalize this well-known and fundamental theory, we plan to simultaneously manipulate multi-level information in both the "what" domain (e.g., tone pitches) and the "when" domain (e.g., tone onset) in future research.

## Methods

### Participants

We recruited 34 participants (17 males and 17 females; age: 23 ± 2.9 years old). The inclusion criteria were: (1) age between 20 and 50 years old; (2) no severe deficit in hearing, eyesight, and color discrimination that could cause problems in understanding experimental procedures; (3) no medical history and diagnosis of neurological or psychological diseases reported by the participant. All participants signed consent forms after understanding the procedure and before the experiment started. The protocol was approved by the ethical committee of the University of Tokyo (No. 21-372) and has been conducted according to the principles expressed in the Declaration of Helsinki. The data from three participants were excluded because they were easily distracted during the experiment or missed structural MRI acquisition (16 males and 15 females; age: 23 ± 3 years old).

### Stimulus

The warning signal was composed of an auditory stimulus (three combined pitches: 350, 700, and 1,400 Hz, 55 dB) and a visual stimulus (a white dot with a 15-pixel diameter placed at the center). The duration of the warning signal was 0.1 s. The target signal was composed of an auditory stimulus (three combined pitches: 500, 1,000, and 1,500 Hz) and a visual stimulus (a white dot with a 15-pixel diameter placed at the center). The duration continued until the *zero* button was pressed or lasted for 1 s if no *zero*-pressed response was detected. The stimulation layout on the monitor (a resolution of 1,920 * 1,080 pixels) included (1) a gray background (RGB: [85, 85, 85]), (2) a fixation (a white cross with a 40-pixel diameter) placed 110 pixels below the center of the monitor. The auditory stimuli were delivered through a pair of desktop speakers. The stimulation was programmed using MATLAB-based Psychtoolbox [34,35] and presented in a dim and sound-proof booth.

### Foreperiod sequence paradigm

We paired two foreperiod trials as a sequence (FP1 and FP2) to establish two regularities: the single foreperiod and the foreperiod sequence. With each sequence, two foreperiod trials were separated by a 1.2-s interval between the offset

of the target in FP1 and the onset of the warning in FP2. Between consecutive sequences, the interval ranged from 3 to 3.2 s, with increments of 0.05 s, between the offset of the target in FP2 and the onset of the warning in the next FP1. To make foreperiod chunking obvious to the participant, a black square (a size of 400 ∗ 400 pixels placed 110 pixels below the center) was presented 0.5 s after the press response for the target in FP2 or 1 s after the onset of the target when the press response was absent. Foreperiod trials within the duration range between 0.4 and 1.1 s and between 1.3 and 2 s are denoted as *S* and *L*, respectively. The participants underwent prior testing to be familiar with foreperiod trials having relatively short or long durations. There were four sequence types: *LL*, *SS*, *LS*, and *SL*, where the first and the second in each pair represent the duration of FP1 and FP2, respectively.

Four blocks were designed, each containing 50 trials of two-foreperiod sequences. In Block 1, there were 25 trials of *LL* and 25 trials of *SS*, leading to an unconditional probability distribution with two nonoverlapping unimodal probability distributions, respectively, peaking at 0.8 and 1.6 s. The 50–50 unconditional probability distribution represents 50% cumulative probabilities in the range of *S* and 50% cumulative probabilities in the range of *L*. For conditional probability distribution, a 0–100 distribution after long FP1 was calculated, representing close to 0% cumulative probabilities in the range of *S* and close to 100% cumulative probabilities in the range of *L*. Similarly, a 100–0 distribution after short FP1 was calculated. In Block 2, there were 25 trials of *LS* and 25 trials of *SL*, resulting in a 50–50 unconditional probability distribution, a 100–0 conditional probability distribution after long FP1, and a 0–100 conditional probability distribution after short FP1. In Block 3, there were 20 trials of *LL*, 5 trials of *LS*, 20 trials of *SS*, and 5 trials of *SL*, resulting in a 50–50 unconditional probability distribution, a 20–80 conditional probability distribution after long FP1, and an 80–20 conditional probability distribution after short FP1. In Block 4, there were 5 trials of *LL*, 20 trials of *LS*, 5 trials of *SS*, and 20 trials of *SL*, resulting in a 50–50 unconditional probability distribution, an 80–20 conditional probability distribution after long FP1, and a 20–80 conditional probability distribution after short FP1. S9 and S10 Figs detail the trial numbers of foreperiod with different durations across four blocks, respectively, for all participants and for one participant. As shown in S10A Fig, we assigned one trial each to foreperiods of 0.4, 0.5, 0.6, 1.0, 1.1, 1.3, 1.4, 1.9, and 2.0 s to maintain a nearly complete and continuous distribution of foreperiods within each block. This design allowed for more accurate estimation of HFs over time while minimally affecting the learning of conditional probabilities. In particular, foreperiods of 1.1 and 1.3 s—which are difficult to clearly categorize as short or long—were included with only one trial each. Their influence was confirmed to be minimal by control analyses excluding these foreperiods (S13 Table), which yielded results similar to the main findings (Table 3).

During the experiment, the participants were instructed to (1) sit comfortably, (2) rest their chins on a chin supporter, (3) look at the fixation to minimize head and eye movements, and (4) press the *zero* button as soon as the target signal appeared. The four blocks were presented in a random order and two subsequent repeats followed. After each block presentation, there was a short rest, and the participants were asked to identify which sequence type appeared most frequently to ensure their engagement with the experiment. Each participant completed a total of 600 FP1 trials and 600 FP2 trials, with the number of trials determined primarily based on prior studies [26,10]. We also confirmed the stability of individual results using bootstrap resampling, which showed that the distribution of individual *R*-squared values was unimodal, with the actual values close to the bootstrap mean and within the 95% confidence intervals (S11 and S12 Figs).

## Data acquisitions

We recorded EEG signals using the 64-channel actiCAP slim from Brain Products and captured individual electrode locations using a 3D camera (brand: STRUCTURE). To reconstruct EEG sources, we also collected individual structural MRI images (T1) using SIEMENS 3T Magnetom Prisma. These two measurements were conducted either on the same day or on two separate days.

**EEG recording.** The 64-channel electrode cap was capped according to anatomical landmarks including nasion, left, and right porions (a top of the ear canal). We captured electrode locations and the three landmarks (labeled with red

stickers) using the 3D camera mounted on an iPad. During EEG recording, event codes were sent at the onset of the warning signal in each foreperiod trial. Raw EEG signals were recorded with a 1000-Hz sampling rate.

**MRI acquisition.** Before image acquisition, the participants completed an MRI safety checklist, and a device was used to detect whether there was any metal material on body. The three anatomical landmarks were labeled using sugar candies (小林製薬のブレスケア) which can be identified in MRI images. The use of the three landmarks allowed us to align electrode locations and MRI images to the same anatomical axis in later analyses. During acquisition, comfortable air-filled paddings were used to prevent severe head motion, and earplugs were used to reduce scanner noise. The T1 acquisition protocol included a field of view: 240×256 mm, 300×320 matrix, TR: 2,400 ms, TE: 2.22 ms, flip angle: 8°, and 0.8-mm slice thickness.

## Data analyses

**Hazard function.** HF, describing the conditional probability of an event occurring at time $t$ given it has not yet occurred, was used to estimate dynamics of temporal predictions. The formula is listed below:

$$\text{HF}(t) = \frac{f(t)}{1 - C(t)} \tag{1}$$

where $t$ represents time points with the range of 0.4–2 s. The function $f$ represents the probability distribution of the foreperiod in a block. The function $C$ represents the cumulative probabilities until time $t$.

When $1 - C(t)$ is close to zero, $\text{HF}(t)$ is toward infinity.

$$\text{HF}(t) = \max\{\text{HF}(0),\ \text{HF}(1),\ ...,\ \text{HF}(t-1)\} \tag{2}$$

Finally, we normalized hazard values between 0 and 1

**Linear mixed-effect analysis.** We excluded only the false alarm trials from our analysis, meaning we did not disregard outlier trials or trials from the early phase of each block. When considering outlier trials, we could, for instance, exclude those with reaction times that are more than 2.5 standard deviations above or below the mean. In this case, there is no significant difference in the number of trials among the four blocks (S5 Table). However, it's crucial to recognize that removing outliers is based on certain assumptions, such as participants not paying attention to these trials. Also, alternative methods, such as excluding trials that fall outside the 25th and 75th percentiles, might yield different outcomes. Regarding trials in the early phase, our analysis found no significant difference in reaction times between the first 20 trials and the last 20 trials (S6 Table). Given the lack of significant differences and our concerns about potential data loss, we decided to keep these outliers and early-phase trials in our linear mixed-effects analysis. While trial-level variability from outliers attenuated the model fit (see the comparisons in S3 Table), we emphasize that the overall conclusions remain unchanged, and the robustness of our results can be further demonstrated by analyses using nearly the entire dataset.

To estimate correlations between hazard values (i.e., temporal prediction) and reaction times, we used linear mixed-effect models while individual differences were controlled. The durations of FP1 ($L$ or $S$) were also controlled when reaction times following FP2 were modeled. The regression models are as below:

For FP1:

$$\text{RT}_t \sim \text{HF}_t + (1|\text{Sub}) \tag{3}$$

For FP2:

$$RT_t \sim \text{HFU}_t + (1|\text{Sub}) + (1|\text{FP1}) \tag{4}$$

$$RT_t \sim HFC_t + (1|Sub) + (1|FP1) \tag{5}$$

$$RT_t \sim HFU_t + HFC_t + (1|Sub) + (1|FP1) \tag{6}$$

$$RT_t \sim HFU_t + HFC_t + HFU_t * HFC_t + (1|Sub) + (1|FP1) \tag{7}$$

where $RT_t$ represents log-transformed reaction times to the target signal appearing at time $t$ (relative to the onset of a warning signal); $HFU_t$ represents an $HF_U$ value at time $t$; $HFC_t$ represents an $HF_C$ value at time $t$.

Additionally, some studies have employed log-transformed reaction times in regression models to demonstrate a negative relationship between log-transformed reaction times and hazard values [9,10]. For completeness, we also conducted analyses using log-transformed reaction times, and our conclusions remained consistent (see S7, S8, and S9 Tables). Moreover, we also performed analyses with HFs based on probability distributions of the actual trial configurations (refer to the example in S10 Fig). The results aligned closely with the model utilizing HFs in Fig 1E (see S10, S11, and S12 Tables and S13, S14 Figs).

The analyses were conducted using R (4.2.2). The functions, *lmer*, *anova*, and *r.squaredGLMM*, were used for the regression analyses, model comparisons, and conditional R-squared, respectively.

**EEG preprocessing.** The raw EEG signals were preprocessed using MATLAB-based EEGLAB for each participant. Firstly, channels showing high amplitudes (over 100 microvoltage) in over than half of the dataset were removed (*pop_select.m*), and signals below 0.1 Hz were filtered out (*pop_eegfiltnew.m*). Independent component analysiswas used for component extraction (*pop_runica.m*), and components containing artifacts were removed using the ADJUST plugin toolbox (*interface_ADJ.m*). Next, the data were segmented into epochs with a time range between −0.8 and 2.8 s relative to the onset of the warning for each foreperiod trial (*pop_epoch.m*), and epochs showing high amplitudes (over 100 microvoltage) were manually removed. The epoch data were then referenced to the average of signals across the 64 channels (*pop_reref.m*), and 1-Hz high-pass and 55-Hz low-pass filters were applied (*pop_eegfiltnew.m*, the default setting: linear noncausal finite impulse function, the forward and backward direction, zero-phase, 25% of the lower passband edge as the transition band). Finally, the data were baseline-corrected (−0.2 and 0 s; *pop_rmbased.m*) and down-sampled to 250 Hz (*pop_resample.m*).

For further analyses, we disregarded trials with the foreperiod lasting < 0.8 s or trials with the false alarm. The processed EEG signals were estimated between 0.4 s after the onset of the warning and the onset of the target.

**Source reconstruction analysis.** The analysis aims to estimate volume conductivity (i.e., forward model) and interpolate sources of electrical potentials (i.e., inverse model) for each participant. First, a head model (also known as a volume conduction model) was created using structural MRI images with the finite element method. The images were resliced into isotropic dimensions ($256 * 256 * 256$ [voxel]) (*ft_volumeslice.m*) and realigned to the three candy-labeled markers (*ft_realign.m*). Then, the images were segmented into five tissue types (gray matter, white matter, cerebral cerebrospinal fluid, skull, and scalp) (*ft_volumesegent.m*). The boundary was adjusted to create a hexahedron for each voxel with the shifting parameter set to 0.3 (*ft_preparemesh.m*). The volume conductivity, [0.33 0.14 1.79 0.01 0.43], was assigned in the order of the above-mentioned tissue types (*ft_prepare_headmodel.m*). Second, the electrode locations captured in the 3D image were aligned according to the three red-dot-labeled markers (*ft_meshrealign.m*), and electrode names were assigned manually (*ft_electrodeplacement.m*). The locations were further adjusted according to the brain shape for better fitness (*ft_plot_headshape.m*; *ft_electroderealign.m*).

Third, the *leadfield* (potential contributions from dipoles to electrodes) was created with the head model and the electrode locations, with a resolution of 1 center (*ft_prepare_leadfield.m*). Then, linear-constraint minimum-variance

beamforming (*lcmv* beamforming) was applied to extract spatial filters (*ft_sourceanalysis.m*). The input products included the processed EEG signals, the head model, and the *leadfield*. The output product was spatial filters (dimensions: *xyz-axes * electrode * source*). Please note that the source numbers varied because of the use of individual MRI images. To acquire source signals in dimensions of *time points* and *source*, spatial filters of each source (*xyz-axes * electrode*) were multiplied by the EEG (*electrode * time points*), resulting in a time course in three axes (*xyz-axes * time points*). We used principal component analysis to extract a time course along a dominant axis (*time points*).

**Multivariate temporal response function analysis (mTRF).** The multivariate temporal response function analysis (mTRF) was used to evaluate correlations between hazard values (i.e., temporal prediction) and EEG source signals. This analysis enables a convoluted regression of a stimulus vector (e.g., $HF_C$) against neural responses (e.g., a time course of a source), and has been widely used in speech research [13] and recently in time research [10]. The formula is as follows.

$$r(t, n) = \sum_{\tau} w(\tau, n)\, s(t - \tau) + \varepsilon(t, n) \tag{8}$$

$$w = (s^T s + \lambda I)^{-1}\, s^T r \tag{9}$$

where $r$ represents the EEG source response for one trial of one participant. $t$ represents a time point in a range between 0.8 and 2 s relative to the cue onset. To match the sampling rate of the source response (in 250 Hz), hazard values (in 10 Hz) were splined-interpolated (*interp1.mat*). $n$ represents an EEG source (~3,000 sources in total). $\tau$ represents a time lag, set to −0.1 and 0.2 s for minimal and maximal time lags with a step of 4 ms as a step. $w$ represents the TRF in the dimension of 76 lags *~3,000 sources. $s$ represents the HF. $\varepsilon$ represents a residual response not explained by the model. $I$ represents the identity matrix. $\lambda$ represents a ridge parameter or the smoothing constant, set to 1. The analysis was conducted using the MATLAB-based mTRF toolbox. We determined the parameters $\lambda$ and $\tau$ based on optimal correlation coefficients between hazard values and EEG signals in the cross-validation (*mTRF-crossval.m*). Specifically, we tested the ridge parameter $\lambda$ across a log-spaced range from $10^{-3}$ to $10^3$ (i.e., 10.^(−3:1:3)) and evaluated four combinations of time lag parameters ($\tau$), with minimum lags set to either −0.2 or −0.1 s and maximum lags to either 0.2 or 0.3 s. In each fold of the outer loop, one trial was held out as the test set, while the remaining $N-1$ trials served as the training set. For each combination of the ridge and time lag parameters (inner loops), a model was trained on the training set and tested on the held-out trial. This procedure was repeated for all $\lambda$ values and time lag combinations (inner loops) across all trials (outer loops), resulting in performance metrics (i.e., correlation coefficients) for each parameter set across all folds. The optimal $\lambda$ and time lag parameters were selected as those that yielded the highest average correlation across the outer folds.

For each participant, the leave-one-out correlation coefficients were evaluated as follows (Fig 3): (1) for each trial, it was temporarily removed from the data; (2) for each source, responses of the remaining ($n$–1) trials were modeled against the corresponding hazard values (*mTRFtrain.m*); (3) the output was the TRF $w$; (4) for each source, the TRF and the HF of the removed trial were used to reconstruct a predicted response of the removed trial; (5) Pearson correlation coefficient between the predicted response and the actual response was estimated for each source (*mTRFpredict.m*). For FP1, the response was modeled against $HF_U$. For FP2, the response was modeled against (1) $HF_U$-only, (2) $HF_C$-only, (3) $HF_U$ and $HF_C$, and (4) both with the interaction term. For each participant, the final correlation coefficients ($n$ trials *~3,000 sources) were averaged across trials. Then, the source data were projected to the voxel level of the brain (7109137 voxels).

Moreover, we also modeled the response against shuffled hazard values as a benchmark control. We shuffled hazard values for each trial and participant. We segmented hazard values into five parts (e.g., 250 values with 50 values per part) and rearranged the order of these parts. This shuffle can contain partial changes of the HF, compared to the total randomization. For example, one shuffle with the order of 2-1-3-5-4 would thus be

(51th–100th)-(1st–50th)-(101st–150th)-(201st–250th)-(151st–200th). For the shuffled HF as controls, we only found significant correlates of shuffled $HF_U$ (S15 Fig) and disregarded them from the correlates shown in Fig 4A.

**Statistics and visualization.** In order to assess whether the correlation coefficients significantly differ from zero, we performed group-level analyses, and the procedure included source interpolation, volume normalization, and source statistics. For each participant, sources, each containing the correlation coefficient, were interpolated back to their individual MRI images (*ft_sourceinterpolate.m*). The anatomical layout was then normalized to the standard brain map, with parameters set to T1.nii in SPM12 and a nonlinear transformation (*ft_volumenormalise.m*). We tested whether the correlation coefficient of each brain voxel is significantly above 0 or significantly different from the one obtained using different variables (e.g., only $HF_U$ versus $HF_U + HF_C$) in the models across the participants. The parameters were set to Monte Carlo method, cluster-based correction, 1,000 randomization, and an alpha level of 0.05 (*ft_sourcestatistics.m)*. Specifically, the Monte Carlo method is a nonparametric randomization approach used to estimate statistical significance by generating a null distribution through repeated permutations of the data. In each permutation, the data from the two conditions (e.g., Model 1 performance versus zero performance) were either left unchanged or sign-flipped within each participant to simulate the null hypothesis. Group-level test statistics (i.e., t-values) were then computed across all voxels for each permutation. To correct for multiple comparisons, we employed the cluster-based maxsum method. In the method, voxel-wise t-values were thresholded at a cluster-forming alpha level of 0.025 (0.05/2 for two-sided testing). Spatially, "adjacent" and "significant" voxels were grouped into clusters, and each cluster was assigned a statistic value equal to the sum of its t-values. Only the largest cluster-level statistic value was kept. We performed 1,000 permutations to form a null distribution of maximum cluster sums. An observed cluster was considered statistically significant if its summed t-value exceeded the threshold percentile of the null distribution (0.05/2 for two-sided testing). Then, the significances were visualized on cortical surfaces of the MNI brain with parameters set to a nearest projection and no lighting (*ft_sourceplot.m*).

## Supporting information

**S1 Text. Supplementary methods for calculating different blurred models.**
(DOCX)

**S1 Table. False alarm trial numbers across blocks.**
(DOCX)

**S2 Table. Average reaction times.**
(DOCX)

**S3 Table. Comparisons of linear mixed-effects models on reaction times (false alarms only vs. false alarms and reaction times outliers excluded).**
(DOCX)

**S4 Table. Average reaction times following FP2.**
(DOCX)

**S5 Table. Outlier trial numbers across four blocks.**
(DOCX)

**S6 Table. Average reaction times across different trials.**
(DOCX)

**S7 Table. Effect of $HF_U$ on log-transformed reaction times following FP1.**
(DOCX)

**S8 Table. Comparisons of linear mixed-effect models on log-transformed reaction times following FP2.**
(DOCX)

**S9 Table. Effects of $HF_U$ and $HF_C$ on log-transformed reaction times following FP2.**
(DOCX)

**S10 Table. Effect of actual $HF_U$ on reaction times following FP1.**
(DOCX)

**S11 Table. Comparisons of linear mixed-effect models on reaction times following FP2.**
(DOCX)

**S12 Table. Effects of actual $HF_U$ and $HF_C$ on reaction times following FP2.**
(DOCX)

**S13 Table. Effects of $HF_U$ and $HF_C$ on reaction times following FP2 (1.1- and 1.3-sec FP removed).**
(DOCX)

**S1 Fig. Average and predicted reaction times across participants.** The average reaction time following FP2 for each block is shown by the black line, with dominant foreperiod durations marked by black circles. The shade represents the standard deviation. Predicted reaction times are shown by the red line, with dominant foreperiod durations marked by red squares. For FP2, predicted reaction times were obtained from Models 1, 2, and 3. The data underlying this figure can be found in https://doi.org/10.17605/OSF.IO/VEDHP.
(EPS)

**S2 Fig. Splined-interpolated hazard values.** The original hazard values at a 10-Hz sampling rate are represented in orange, while the interpolated values at a 250-Hz sampling rate are represented in blue. The data underlying this figure can be found in https://doi.org/10.17605/OSF.IO/VEDHP.
(EPS)

**S3 Fig. Temporal response function of $HF_U$.** Within the significant area for $HF_U$-only in Fig 4A, the positive and negative average TRF (i.e., weight) between source responses and $HF_U$ values at a 0-s lag is shown. Values were normalized between −1 and 1. The data underlying this figure can be found in https://doi.org/10.17605/OSF.IO/VEDHP.
(EPS)

**S4 Fig. Temporal response function of $HF_C$.** Within the significant area for $HF_C$-only in Fig 4B, the positive and negative average TRF between source responses and $HF_C$ values at a 0-s lag is shown. Values were normalized between −1 and 1. The data underlying this figure can be found in https://doi.org/10.17605/OSF.IO/VEDHP.
(EPS)

**S5 Fig. Temporal response functions of $HF_U$, HFC, and their interaction.** Within the significant area for $HF_U++HF_C++HF_U*HF_C$ in Fig 4C, the positive and negative average TRFs between source responses and each of those three values at a 0-s lag is shown. Values were normalized between −1 and 1. The data underlying this figure can be found in https://doi.org/10.17605/OSF.IO/VEDHP.
(EPS)

**S6 Fig. Event-related potentials (ERP) during FP1 and FP2.** We averaged processed EEG signals in foreperiod trials lasting >0.7 s. Time zero represents the onset of the warning signal. Differences between ERPs during FP1 and FP2 (FP1 − FP2) were tested using the Monte Carlo method and cluster-based correction method (1,000 randomization, a two-tailed test, and an alpha level of 0.05). Triangles in the topography and gray shades in the time

course represent the significance. The data underlying this figure can be found in https://doi.org/10.17605/OSF.IO/VEDHP.
(EPS)

**S7 Fig. R-squared obtained using different representations of the temporal predictions.** The probability distribution is denoted as PDF, the hazard function as HF, temporally blurred as temp-blurred, and probabilistically blurred as prob-blurred. **(A)** The adjusted $R$-squared was obtained from each participant for FP1, and the conditional $R$-squared was obtained (with FP1 duration as a random effect). The black circle represents the mean of the R-squared across participants. **(B)** The mean R-squared was obtained across participants across blurring parameters ranging from 0.15 to 0.35 in steps of 0.1. The results on the left and right columns were obtained from the regression model formulas (3) and (7), respectively. The data underlying this figure can be found in https://doi.org/10.17605/OSF.IO/VEDHP.
(EPS)

**S8 Fig. Average reaction times following FP2.** The visualization of S3 Table. The data underlying this figure can be found in https://doi.org/10.17605/OSF.IO/VEDHP.
(EPS)

**S9 Fig. Probability distributions of the actual trial configuration averaged across participants.** The red line represents the hazard function derived from the probability distribution function in Fig 1D. The data underlying this figure can be found in https://doi.org/10.17605/OSF.IO/VEDHP.
(EPS)

**S10 Fig. The actual trial configuration from one participant.** There were 50 trials of the two-foreperiod sequence, resulting in a total of 100 foreperiod trials for each block **(A)** Initial trial numbers for each participant and block were determined based on a 50−50 probability distribution. **(B)** For each participant, foreperiods were randomly selected in the ranges of $L$ and $S$, and paired as $LL$, $SS$, $LS$, or $SL$. An example of actual trial numbers for one participant is shown. The data underlying this figure can be found in https://doi.org/10.17605/OSF.IO/VEDHP.
(EPS)

**S11 Fig. Bootstrap distribution of adjusted R-squared values based on 1,000 resamples of reaction times following FP1 for each participant.** Reaction times were regressed against $HF_U$ in the linear regression model. The blue line represents the actual adjusted $R$-squared value; the green line represents the mean of the bootstrap distribution; and the red dashed lines represent the 95% confidence interval. The data underlying this figure can be found in https://doi.org/10.17605/OSF.IO/VEDHP.
(EPS)

**S12 Fig. Bootstrap distribution of conditional R-squared values based on 1,000 resamples of reaction times following FP2 for each participant.** Reaction times were regressed against $HF_U$, $HF_U$, and the interaction term in the linear mixed effect model, with the FP1 duration as a random effect. The representation is the same as S11 Fig. The data underlying this figure can be found in https://doi.org/10.17605/OSF.IO/VEDHP.
(EPS)

**S13 Fig. _R_-squared obtained using hazard functions from the actual trial configuration.** The representations are the same as those in S7A Fig. The results in the left and right columns were obtained from the regression model formulas (3) and (7), respectively. The data underlying this figure can be found in https://doi.org/10.17605/OSF.IO/VEDHP.
(EPS)

**S14 Fig. Average and predicted reaction times across participants.** The same representations as Fig 2 are used. The data underlying this figure can be found in https://doi.org/10.17605/OSF.IO/VEDHP. (EPS)

**S15 Fig. Neural correlates of original and shuffled $HF_U$.** EEG source responses during FP1 were modeled against shuffled $HF_U$ as a control in the panel **A** and original $HF_U$ in the panel **B**. Four cortical surfaces with significant correlation coefficients are shown. The color bar shows correlation coefficients. The data underlying this figure can be found in https://doi.org/10.17605/OSF.IO/VEDHP. (EPS)

## Acknowledgments

We thank Lu Li and Junko Taniai for helping with participant recruitment and experiment preparation. We also thank Felix B. Kern for proofreading.

## Author contributions

**Conceptualization:** Yiyuan Teresa Huang, Zenas C. Chao.

**Data curation:** Yiyuan Teresa Huang.

**Formal analysis:** Yiyuan Teresa Huang.

**Funding acquisition:** Yiyuan Teresa Huang, Zenas C. Chao.

**Investigation:** Yiyuan Teresa Huang, Zenas C. Chao.

**Methodology:** Yiyuan Teresa Huang.

**Project administration:** Yiyuan Teresa Huang, Zenas C. Chao.

**Supervision:** Zenas C. Chao.

**Visualization:** Yiyuan Teresa Huang.

**Writing – original draft:** Yiyuan Teresa Huang.

**Writing – review & editing:** Zenas C. Chao.

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
