## [Editor Report · Decision Letter 0]

30 Jun 2025

Dear Dr Huang,

Thank you for submitting your manuscript entitled "Temporal Prediction through Integration of Probability Distributions of Event Timings at Multiple Levels" for consideration as a Research Article by PLOS Biology.

Your manuscript has now been evaluated by the PLOS Biology editorial staff as well as by an academic editor with relevant expertise and I am writing to let you know that we would like to send your submission out for external peer review.

Once your full submission is complete, your paper will undergo a series of checks in preparation for peer review. After your manuscript has passed the checks it will be sent out for review. To provide the metadata for your submission, please Login to Editorial Manager (https://www.editorialmanager.com/pbiology) within two working days, i.e. by Jul 02 2025 11:59PM.

Kind regards,

Christian

Christian Schnell, PhD

Senior Editor

PLOS Biology

cschnell@plos.org

---

## [Decision Letter · Decision Letter 1]

17 Sep 2025

Dear Dr Huang,

Thank you for your patience while we considered your revised manuscript "Temporal Prediction through Integration of Probability Distributions of Event Timings at Multiple Levels" for publication as a Research Article at PLOS Biology. This revised version of your manuscript has been evaluated by the PLOS Biology editors, the Academic Editor and the original reviewers.

Based on the reviews and on our Academic Editor's assessment of your revision, we are likely to accept this manuscript for publication, provided you satisfactorily address the following data and other policy-related requests. We note that Reviewer 1 still has a large number of concerns and is not convinced that your model provides a good fit for the experimental data. After discussing these concerns with the other reviewers and the Academic Editor, we think that these are important concerns and do not dismiss them, but still think that your study provides a valuable contribution to the literature, noting that further studies are necessary.

* We would like to suggest a different title to improve its accessibility for our broad audience:

"Humans integrate both unconditional and conditional timing expectations into the timing of their behavioral response"

* Please add the links to the funding agencies in the Financial Disclosure statement in the manuscript details.

* Please include information in the Methods section whether the study has been conducted according to the principles expressed in the Declaration of Helsinki.

* DATA POLICY:

Regardless of the method selected, please ensure that you provide the individual numerical values that underlie the summary data displayed in the following figure panels as they are essential for readers to assess your analysis and to reproduce it: 4D, S7A and S13.

* CODE POLICY

We expect to receive your revised manuscript within two weeks.

*Published Peer Review History*

*Press*

Sincerely,

Christian

Christian Schnell, PhD

Senior Editor

cschnell@plos.org

PLOS Biology

Reviewer remarks:

Reviewer #1: ===========================================================================

REVISION ROUND 2 - COMMENT REVIEWER #1:

The authors have replied to all of my comments in writing and provided several

additional analyses. To facilitate easy reference to the previous round of review, I briefly reply

to the authors in a comment-by-comment fashion below. But first, let me preface my review with an important aspect of the modeling work that has not yet been resolved in the revised

manuscript.

Preface

The revised manuscript does not offer convincing evidence that the hazard-function-based models provide an adequate fit to the reaction time data over the range of foreperiods.

Quantitatively, the goodness-of-fit is weak (numerically small adj. R2: FP1: 0.173, Table 1;

FP2: 0.215 to 0.221, Table 2). Also the beta values are very small (-0.008 to -0.049) indicating

weak correlation. Suppl. Fig. 2 depicts this weak correlation.

REVISION ROUND 2 - AUTHORS' RESPONSE:

We thank the reviewer for the careful evaluation and for highlighting important issues regarding the model fit and interpretation of our results. Before addressing specific concerns, we would like to clarify a terminology issue. In the original manuscript, we mistakenly referred to the model fit metric as "adjusted R2". However, since we used a linear mixed-effects model, the appropriate metric is "conditional R2", which accounts for both fixed and random effects. Importantly, this was a labeling error only; the calculations and reported values are correct and remain unchanged. We have revised the terminology throughout the manuscript and supplementary materials to accurately reflect this correction. Below, we respond point by point to the comments.

Response 1: Model fit

We acknowledge the reviewer's concern regarding the modest conditional R2 values (FP1: 0.173; FP2: 0.215-0.221). These values were obtained from analyses in which only false alarm trials were excluded, while outliers and the first 20 trials of each block were retained. This decision was made in response to the previous Comment 4 and reflects our approach to minimize assumptions about the underlying processes of outlier trials.

To examine whether trial-level noise from outliers contributed to the modest fit, we conducted a follow-up analysis using a stricter dataset in which both false alarms and RT outliers (defined as ±2.5 SD) were excluded. This resulted in consistently improved conditional R2 values (see details below):

* FP1: Conditional R2 increased to 0.262

* FP2: Conditional R2 increased to 0.295-0.304

These updated results are now reported in Supplementary Table 3. The main text has also been revised to indicate that trial-level variability attenuated the explanation of the relationship between HF and RT in our original analysis. Additionally, we tested model fit using log- transformed RTs and obtained conditional R2 values of 0.301 for FP1 and 0.340-0.353 for FP2. However, in the main text, we chose to retain the original data selection and analysis approach, which excludes only false alarms and uses non-transformed RTs. This decision was based on two considerations: first, to avoid introducing additional assumptions about the underlying processes of outliers and the assumption of normally distributed reaction times; and second, although the R2 values are not high, they are reasonable compared to those reported in existing publications, as we will discuss below.

-----SUPPL TABLE 3

Comparison with previous studies

Our study yielded modest conditional R2 values, but they are reasonable when compared to prior studies, particularly considering: (1) the use of a more complex probability distribution structure, (2) the inclusion of a full range of foreperiods, and (3) the introduction of both unconditional and conditional probability regularities within a single task. We elaborate on each of these below and summarize them in the table that follows.

REVISION ROUND 3 - COMMENT REVIEWER #1:

Before we get lost in the argumentative weeds here: Even after sub-selecting data, the fits remain quantitatively weak (R-squared <= 0.3). For the best-fitting model of behavior of the manuscript, these values are not "reasonable" at all.

In the table that compares R-squared across different studies, the authors provide a selection of four published studies. Two of these studies report R-squared values between 0.429 and 0.53 (Herbst 2018, Bueti 2010). Obviously, the authors included these two studies to argue that the small R-squares values observed in their own work are ok. First, these two studies offer better fits to data with substantially larger R-squared than the current manuscript. Second, the standard for what is a "reasonable" fit of HF to RT is much higher than the authors want to make us believe here by including these two papers. In the temporal prediction literature there are many papers that offer much better HF model fits to RT with larger R-squared. E.g.:

Pasquereau et al. J Neurophysiol 2015: Fig. 2e&f, R-squared = [0.82, 0.97]

Sharma et al. Cereb Cortex 2015: Fig 2b, R-squared = 0.972

Tsunoda et al. Neurosci Lett, 2008: Fig 4B, R-squared = [0.91, 0.93]

In the following sections, the authors try to explain why their HF fits are weak. It appears that the authors make a crucial mistake here: The question is not "Why does the hazard function not fit the data well?", as the authors seem to think, but rather: "What computational model fits the data really well?".

The very weak quantitative evidence for the hazard function hypothesis reported here does not justify the use of the model as a regressor on neural data.

REVISION ROUND 2 - AUTHORS' RESPONSE:

---TABLE

1. Studies using simple probability distributions yield better fit

For instance, Herbst et al. (2018) applied a linear mixed-effects model using log-transformed RTs and HF as a fixed effect, reporting conditional R2 values of 0.429-0.448 for unimodal and uniform distributions, respectively. On the other hand, Grabenhorst et al. (2019) reported even higher model fits (adjusted R2 values approaching 1; see their Supplementary Figure 13) using exponential and flipped distributions without catch trials.

Although these studies reported higher R2 values, a key difference lies in the simplicity of their probability structures, which typically produce hazard functions or probability distributions with a single peak or monotonic increase used for model fitting. In contrast, our bimodal distributions generate dual expectancy peaks, resulting in more complex temporal expectations that cannot be adequately captured, such as, by a simple monotonic drift in brain signals.

REVISION ROUND 3 - COMMENT REVIEWER #1:

The authors argue that the small R-squared values of their models result from the higher complexity of their task. This argument does not logically support their model choice. Whatever the complexity of the task, an adequate model needs to capture the behavior, resulting in larger R-squared. If the HF does not fit the data well, then it is not a good model.

REVISION ROUND 2 - AUTHORS' RESPONSE:

2. Design issues in studies using bimodal probability distributions

Some studies have reported high R2 values with bimodal distributions. For example, Janssen and Shadlen (2005) reported values of 0.95 and 0.96 in two non-human primates, and Bueti and Macaluso (2010) reported a value of 0.53 in humans. However, these studies did not include a full range of foreperiods. In Janssen and Shadlen (2005), for instance, the bimodal distribution had peaks around 0.27 and 1.93 seconds, but no trials were presented between 0.75 and 1.75 seconds (i.e., between the peaks). There is the omission of intermediate foreperiods with low probability, where presumably few trials are set and high uncertainty could be caused. This design reduces trial-by-trial variability and likely improves model fit by this omission.

In contrast, our design incorporated a nearly continuous range of foreperiods (excluding only 1.2 seconds to avoid categorical confusion between short and long FPs). This allowed us to capture the full dynamics of the hazard function, providing a more comprehensive and naturalistic test of temporal prediction over time. However, this approach also introduced greater variability, which can reduce R2. Notably, this comprehensive design was not limited to our study but also adopted in the above studies using a simpler probability structure.

REVISION ROUND 3 - COMMENT REVIEWER #1:

The autors state that their design incorporates a full range of foreperiods and they argue that this introduces greater variability [in RT]. The authors again attribute their weak model fit to differences between other HF papers' tasks and their own task. I can only repeat myself: If the HF does not fit the data well, then it is not a good model. An adequate model needs to better capture the behavior.

REVISION ROUND 2 - AUTHORS' RESPONSE:

3. Task complexity

Finally and most critically, our study is the first to combine two probability distributions within a single task. Moreover, while employing a bimodal distribution, we also varied its structure across blocks (e.g., equal 50-50 peak contributions vs. asymmetric 20-80 peaks). This manipulation was essential for disentangling unconditional and conditional probability structures and also for assessing how the brain generalizes across different probabilistic rules.

Such complexity can increase trial-by-trial (and also inter-individual variability) in learning and strategy, which in turn can reduce the explained variance in model fitting. Still, we would like to emphasize that despite this added complexity, our key behavioral and neural findings remain robust and statistically significant. All major results, including significant behavioral and neural correlations with HFU, HFC, and the interaction term, were consistently verified and shown through control and additional analyses.

We have included the Supplementary Table 3 in the revised Methods:

"While trial-level variability from outliers attenuated the model fit (see the comparisons in Supplementary Table 3), we emphasize that the overall conclusions remain unchanged, and the robustness of our results can be further demonstrated by analyses using nearly the entire dataset." (Page 30, Lines 15-18)

We have also discussed this in the revised Discussion:

"We also acknowledge that the conditional R2 values in our behavioral data appear modest, and the relatively poor model fits can be attributed to several factors. First, we included reaction time outliers and used non-log-transformed data in our analysis (see details in Methods) to avoid introducing additional assumptions. We verified the contribution of trial-level noise from outliers to the model fit by demonstrating consistently improved conditional R2 values when outliers were removed (Supplementary Table 3). Second, unlike previous studies that used exponential or unimodal probability distributions, typically resulting in a single peak or a monotonic increase, we employed bimodal distributions that generate dual expectancy peaks. This probability structure produces more complex temporal expectations that cannot be adequately captured by a simple monotonic drift in brain signals. Third, to ensure a complete probability distribution structure, we included trials across the full foreperiod range in both our design and analysis, even though some foreperiod durations were in few trial numbers, likely introducing trial noise. This allowed us to more comprehensively capture the evolution of the hazard function over time. Finally, our design is the first to create two probability regularities, and to achieve it, we also designed various bimodal distributions across blocks (e.g., equal 50-50 peak contributions vs. asymmetric 20-80 peaks). Such complexity might increase trial-by-trial (also inter-individual) variability in learning and strategy, which in turn can increase behavioral noise and reduce the explained variance in models. Still, we would like to emphasize that despite this added complexity, our key behavioral and neural findings remain robust. All major results, such as both significant behavioral and neural correlates with HFU, HFC, and the interaction term, were consistently verified and shown through control and additional analyses." (Pages 21-22 Lines 16-13)

REVISION ROUND 3 - COMMENT REVIEWER #1:

The authors claim: "our key behavioral and neural findings remain robust and statistically significant." I disagree with this assessment. The behavioral results are obviously not robust (yet again: R-squared < 0.3, systematic deviations between data and model, more on that later). The neural findings hinge entirely on this weak modeling of behavior and are thus also far from being robust. Here a few examples of much more robust behavioral modeling that is then related to neural data:

Janssen & Shadlen Nat Neurosci 2005: Figs. 2a&b, R-squared = 0.77 to 0.96

Pasquereau et al. J Neurophysiol 2015: Fig. 2e&f, R-squared = [0.82, 0.97]

Sharma et al. Cereb Cortex 2015: Fig 2b, R-squared = 0.972

I can only repeat myself: If the HF does not fit the data well, then it is not a good model. An adequate model needs to better capture the RT behavior, irrespective of the task complexity. In this manuscript, the authors developed and used a specific, complex task. The authors should also have developed a specific computational model from first principles, i.e. a model that reflects the computational demands of the task. But the authors opted for fitting an off-the-shelf HF to the data and now they are trying their best to argue that it is a good model.

REVISION ROUND 2 - AUTHORS' RESPONSE:

Response 2: Beta coefficients

The reported beta values were unstandardized and therefore influenced by the scale of the predictor variables. To improve interpretability, we have added standardized beta coefficients, which range from 0.04 to 0.21 across models. While modest in magnitude, these effects are statistically significant and consistent across models and analyses. The standardized beta values have been incorporated into Tables 1 and 3 (also shown below) as well as Supplementary Tables 7, 9, 10, and 12.

REVISION ROUND 3 - COMMENT REVIEWER #1:

The authors acknowledge that their models' beta values are numerically "modest in magnitude". In other words: they are very small. Arguing from the standpoint of statistical significance is not convincing here. There will be many other models that yield such small, yet significant, correlation beta values (especially in light of the large data set which facilitates high statistical power...). Therefore models of RT over foreperiod are commonly assessed based on R-squared and on visual inspection of the fit according to commonly accepted criteria (more on that below).

REVISION ROUND 2 - AUTHORS' RESPONSE:

---TABLE 1

---TABLE 3

REVISION ROUND 2 - COMMENT REVIEWER #1:

Qualitatively, the RT modulation (Suppl. Fig. 1) does not resemble the dynamics of the

(mirrored) HF variables (Fig 1E).

Since all of the manuscript's main results hinge on the HF fits to RT, more must be done to

convince the reader that the HF is indeed an appropriate model of RT (and a neurobiologically

relevant computation).

First, it is crucial for the reader to see the average RT over foreperiod (group-level) plots in the

manuscript (not in the Supplement) so that the reader can assess the modulation of RT.

Second, these average RT over foreperiod (group-level) plots need to show the dynamics of the RT modulation in a clear way. The current Suppl. Fig. 1 does not do this sufficiently: The y-axis spans close to 1 s, while the RT modulation spans less than 0.1 s, making it hard to see any

pattern. All the boxes, error bars, outlier dots etc. in the plots further obscure a clear view of the RT modulation.

The authors need to plot average RT over foreperiod and rescale the y-axis so that the RT

modulation over foreperiods becomes clearly visible. See e.g. these papers for intuitive average RT over foreperiod plots: Janssen & Shadlen, Nat Neuro 2005, Bueti et. al. J Neurosci 2010, Grabenhorst et al. Nat Comms 2019, Schoffelen et al., Science 2005.

Third, Suppl. Fig. 2 shows RT over hazard. This figure does not show the fits *over time*, i.e.

over foreperiods. Instead, in Suppl. Fig. 2, data are collapsed across foreperiods. It is therefore

not possible to visually inspect the model fits to data *over time* (i.e. over foreperiod). Such an assessment is crucial since a convincing model needs to capture RT dynamics over the range of foreperiods. The average RT over foreperiod plots are the common way to display fits to data in these types of experiments (e.g. Fig. 2 in Janssen & Shadlen, Nat Neuro 2007, which the authors cite).

The authors need to provide plots that show how the HR model fits *group-level* average RT

*over time*, i.e. over foreperiods. This should be done for *all models* so that the reader can

assess the choice of models based on visual inspection and not only based on fit metrics (adj. R2, AIC, BIC).

[n.b. The authors say that it is challenging to visualize the interaction term in one of their models (their resply to comment 5). Some ideas: 3D surface plots, line plots across time for different levels of HFX(t), contour plots...)

REVISION ROUND 2 - AUTHORS' RESPONSE:

Response 3: Visualizing average RT over foreperiods and model fit across all models

We fully agree with the reviewer that the original Supplementary Figure 1 did not clearly demonstrate RT changes due to its wide y-axis range and overly detailed plotting elements (e.g., boxplots and outliers). In response, we have created a new Figure 2 that shows group-level average RTs over foreperiods in FP1 and FP2 across all four blocks, using a more appropriate y- axis range and simplified plotting (mean with standard deviation shading only). This figure also includes: (1) the predicted RT values following FP1 from Model 1 (HFU only), (2) the predicted values following FP2 from Model 4 (HFU, HFC, and the interaction term), and (3) the actual trial distribution across all foreperiods at the bottom.

REVISION ROUND 3 - COMMENT REVIEWER #1:

The authors now provide plots in which the y-axis scaling is scaled to accomodate the large RT variance.

1) Noone has asked for plots of variance. This, again, makes it unnecessarily hard to assess model fit and RT modulation: Modulation of RT is much less than 100 ms on most plots, yet y-axis spans ~170 ms.

2) Noone has asked to mark "the foreperiods with dominant trial frequencies" with dots. This just distracts from inspecting the fits. And it is utterly nonsensical: A model of RT over foreperiod needs to fit all the data, not only the foreperiods with dominant trial frequencies.

At this point, after having asked twice for these simple plots, which should have been part of the manuscript from the initial submission, in line with the literature that the authors cite, the authors' plotting strategy seems intended to obscure the weak model fits.

REVISION ROUND 2 - AUTHORS' RESPONSE:

We elaborate further on them in Response 4 below.

We sincerely thank the reviewer for this suggestion, which helped us overcome the challenge of visualizing Model 4 (HFU, HFC, and the interaction term). Accordingly, we have also revised the visualization of model fits from other models in Supplementary Figure 1 and used actual HF values in Supplementary Figure 14.

---FIG 2

---SUPPL FIG 1

Response 4: Model fit interpretations

We agree with the reviewer that both visual inspection and statistical criteria (e.g., AIC, BIC) are critical for evaluating model performance. Several key points should be emphasized regarding the interpretation of model fits in the new Figure 2.

First and also before discussing the model fit in detail, it is important to note that we performed the linear mixed-effects model analysis using data from all four blocks combined. Accordingly, the predicted and actual RTs across all blocks should be considered jointly when evaluating model fit. For example, inspecting the model fit for FP2 involves considering the eight actual and predicted RT curves across the various blocks together.

Second, we note that the predicted RT values (red and orange) generally fall within the range of the observed group mean (black) ± 1 standard deviation (shading), indicating an overall reasonable fit.

REVISION ROUND 3 - COMMENT REVIEWER #1:

The authors argue that a model is reasonable if it predicts with a precision of 1 std of the mean of the to-be-predicted variable. This is definitely not an appropriate definition of a reasonable model fit in RT models in temporal prediction and it appears that the authors are making up their own rules by now.

REVISION ROUND 2 - AUTHORS' RESPONSE:

However, some discrepancies in the trend can be observed, particularly at short foreperiods (e.g., 0.4-0.6 s), where the actual RTs show a drop (black), while the predicted RTs remain relatively flat (red and orange). Several points are worth noting in this context:

1. The predicted values in these early foreperiods are flat because they were associated with very low probability values, deliberately designed to approach zero. As a result, the derived HF values (both blurred and unblurred) were also close to zero, producing flat predicted RTs.

REVISION ROUND 3 - COMMENT REVIEWER #1:

Fig 2 shows that there is a systematic deviation between data and model at short foreperiods. This means that the model is not doing a good job at predicting the data. Which means that the authors' computational assumtions about the brain are wrong.

REVISION ROUND 2 - AUTHORS' RESPONSE:

2. As discussed in Response 1, although these short foreperiod trials were few in number, they were retained to preserve the full hazard function change over time. This approach naturally led to increased variability, reflected in the wider standard deviation shading in those areas.

REVISION ROUND 3 - COMMENT REVIEWER #1:

This argument does not support the weak model fits. Of cause one needs to keep the low probability trials in the analysis. They are part of the probability distribution that the brain has to learn. (n.b. in simple RT tasks average RT and variance of RT are positively correlated. This also holds in probability-based designs and is to be expected.)

REVISION ROUND 2 - AUTHORS' RESPONSE:

3. A similar pattern, where RTs vary while HF remains stable, has also been observed in previous work using longer-tailed probability distributions with sparse trials at both ends, such as Herbst et al. (2018).

REVISION ROUND 3 - COMMENT REVIEWER #1:

The authors argue that their model fit is ok because a phenomenon that their model cannot capture was also not captured by some other model in some other paper. Wouldn't it be more scientific, to model this phenomenon and thus offer a quantitative account? ===========================================================================

REVISION ROUND 2 - AUTHORS' RESPONSE:

To aid visual interpretation, we marked the foreperiods with dominant trial frequencies (0.7, 0.8, 0.9, 1.5, 1.6, and 1.7 s) using dots. These highlighted points more clearly demonstrate that the observed RTs correspond well with both the predicted trend and the peaks in the foreperiod distribution.

REVISION ROUND 3 - COMMENT REVIEWER #1:

Here the authors marked RT and model with dots at the high-probability-foreperiods. They argue that the model fits the data at these foreperiods. I disagree with this statement because this is obviously not the case. As one can see, the RT dots do not correspond well with the model dots. There are systematic differences between the "RT dots" and the "model dots" (more on that below).

In probabilistic designs, such as the one used here, the probability varies across foreperiods (uniform distribution being the exception). A convincing model needs to fit the data over the ENTIRE RANGE of foreperiods. Note that this is a core assumption of the Hazard Function hypothesis in temporal prediction: The brain computes the continuous HF variable across the entire range of foreperiods.

REVISION ROUND 2 - AUTHORS' RESPONSE:

Finally, although the model fits for FP2 in Figure 2 and Supplementary Figure 1 appear visually comparable across the four models, statistical comparisons using AIC and BIC nonetheless identified Model 4 (which includes the interaction term) as the best-fitting model.

REVISION ROUND 3 - COMMENT REVIEWER #1:

As I had already written, model comparison based on AIC/BIC can only be interpreted in *relative* terms. This relative comparison does not offer support for the hazard function as an adequate model at all, it merely indicates that among the compared models, one is the "best". An *absolute* assessment of the model needs to rely on R-squared and beta (both weak) and on visual expection of the fits (also weak, see below).

Fig. 2 is a central data figure of the manuscript. But in their manuscript, the authors do not describe the dynamics of their main RT variable to the reader. It is an unconventional choice to not at all verbally assess how the experimental task shapes behavior and, instead, go straight to the modeling. But it shows how little the authors want to highlight their RT data. Here is an assessment of RT dynamics and how the models relate to it:

On the data:

- Over short foreperiods (< ~1.2s), RT decreases in all blocks, in "FP1" and "FP2" (block 3, rightmost plot deviates from this pattern to some degree).

- Over long foreperiods (> ~1.2s), in all of the blocks, the difference in RT across the range of foreperiods is numerically very small, spanning only ~0.025s (block 3 rightmost plot and block 4 middle plot deviate from this range since each has one strong outlier RT value). This is a very small RT modulation compared to the temporal prediction RT literature. Also, there are no clear patterns of RT modulation, so maybe we are only looking at random RT. This raises the question whether there are any causal effects of the task's probability manipulation on RT over long foreperiods at all.

On the modeling:

FP1 plots (HFu model):

- In blocks 1 to 4, RT decreases over short foreperiods (< ~1.2s). The model predicts a close to constant RT and does not capture the observed modulation.

- Across blocks 1 to 4, over long foreperiods (> ~1.2s), the model predicts a triphasic pattern of RT ("down-up-down"). This pattern is not observed in RT in any of the blocks and thus the model does not predict the data well. In all blocks the 3 HFu model "dots" predict a steeply decreasing RT which does not capture well the corresponding RT "dots" which either decrease with a much shallower slope (blocks 1 to 3) or show a biphasic pattern ("up-then-down", block 4).

FP2 plots (HFc + HFu*HFc model):

- Over short foreperiods (< ~1.2s), the model does not capture the decrease in RT well. It over-estimates the data in block 1, without capturing the decrease. In block 2, it predicts a close-to-constant RT, whereas RT decreases. In block 3 (middle plot), it under-estimates RT and does not capture its decrease. In block 3 (right plot), the model captures the data better. In block 4 (middle plot), it under-estimates the RT data. In block 4 (right plot), it does not capture the range of RT modulation.

- Over long foreperiods (> ~1.2s), the model predicts either a triphasic "down-up-down" (Blocks 2 & 3) or a tetraphasic "up-down-up-down" (Blocks 1 & 4) RT dynamic over a foreperiod range of ~0.8 s. The data do not justify this modeling assumption. In light of the miniscule difference in RT across the range of foreperiods (~0.025 s), such a polyphasic model is inadequate, at best capturing random noise by chance.

When considered jointly across all blocks, the HFc + HFu*HFc model does not capture the data well.

REVISION ROUND 2 - AUTHORS' RESPONSE:

Accordingly, we have revised the Results section.

"We also visualized the observed and predicted reaction times across all foreperiods (Figure 2A). A general trend consistent with the HF can be observed, particularly in the foreperiods with dominant trial frequencies, where actual and predicted reaction times closely align. Note that data from all blocks were combined for the model analysis; however, we separated them in the figure for clearer illustration of the actual reaction times across foreperiods alongside the predicted values." (Page 9, Lines 4-10)

"We first visualized the actual and predicted reaction times for all models in Figure 2 and Supplementary Figure 1. Due to the varied block-wise probability distributions, visually inspecting model fit is more challenging, as it requires jointly evaluating all predicted curves across the four blocks. To statistically determine the best-fitting model, we performed model comparisons." (Page 10, Lines 2-6)

REVISION ROUND 2 - COMMENT REVIEWER #1:

Comment 1

The authors improved the terminology in their manuscript. This is very helpful, as is the table

showing the changes. One minor point: In the intro (P3, l14) the authors say that the hazard rate itself is a conditional probability. While this makes sense in itself, it may throw off the reader who may confuse it with the use of the terms (un-)conditional, as in e.g. Fig. 1D. Maybe rephrase the sentence in the intro to not use the word conditional?

REVISION ROUND 2 - AUTHORS' RESPONSE:

We thank the reviewer for pointing this out. To avoid potential confusion with the terminology used later in the manuscript, we have revised the sentence as follows: "This function, derived from the probability distribution of the foreperiod, describes how the probability of the target signal occurring is updated over time, given that it has not yet occurred." (Page 3, Line 13-15)

REVISION ROUND 3 - RESPONSE REVIEWER #1:

Ok.

REVISION ROUND 2 - COMMENT REVIEWER #1:

The authors performed new model fits to non-log-transformed RT. This is in line with the bulk of the hazard rate literature and does not add neuro-computational assumptions as the log-

transform did. Both improves interpretability. The authors kept the fits to log-transformed RT

and added them to Supplement.

REVISION ROUND 2 - AUTHORS' RESPONSE:

We appreciate the reviewer's understanding on this point.

REVISION ROUND 2 - COMMENT REVIEWER #1

The authors reply to my comment that they regressed HF on EEG data, not RT. I am aware of

this fact. The crucial point is: if the HF fits to RT are not convincing, this does call into question

the HF as an appropriate regressor and thus questions the EEG analysis and the interpretation

of the results, see Preface.

REVISION ROUND 2 - AUTHORS' RESPONSE:

We thank the reviewer for pointing this out. We have addressed this comment by providing an improved visualization of the HF across all foreperiods along with the predicted values in Figure 2 and Supplementary Figure 1 (see Responses 3 and 4). We also show significant and moderate correlations between RT and HF (Response 2), and discuss prior literature that supports the use of HF as a regressor (Response 5 below).

REVISION ROUND 3 - COMMENT REVIEWER #1:

Again: a "moderate correlation" between HF and RT does not justify a regression on neural data. At this point the manuscript has reached a dead end: The authors have not formally developed a model tailored to their task's computational demands but tacitly assumed the Hazard Function's validity in their specific context. But the HF model does not fit the data well and there are no better acounts of the behavior in the manuscript. Frankly, the autors still owe the reader a convincing computational account of the RT data.

REVISION ROUND 2 - COMMENT REVIEWER #1:

Comment 2

Please see Preface above.

REVISION ROUND 2 - AUTHORS' RESPONSE:

We have addressed this comment by providing Figure 2 and Supplementary Figure 1 (see Responses 3 and 4).

REVISION ROUND 2 - COMMENT REVIEWER #1:

Comment 3

The authors convincingly addressed this comment.

Comment 4

The authors convincingly addressed this comment.

Comment 5

i) On the model selection: Please see Preface.

REVISION ROUND 2 - AUTHORS' RESPONSE:

We have addressed the comment above in Responses 3 and 4.

REVISION ROUND 2 - COMMENT REVIEWER #1:

ii) On the temporal uncertainty:

The authors tested temporally and probabilistically blurred HF models and find the results

inconclusive.

Based on the authors' reply to my comment, I get the impression that they think that the different blurring regimes are just a means to improve model fit ("The winner in RT following FP1 is using probabilistically blurred HF. However, the winner in RT following FP2 is using temporally blurred and probabilistically blurred HF." and "However, the optimal blurring effect

varies, depending on different blurring factors").

Instead, these blurring regimes serve to test specific hypotheses about the brain's uncertainty in time estimation.

It is widely accepted that there are errors to neural temporal estimates, e.g. in interval timing.

These timing mechanisms are crucial for sucessful performance of the timing task that the

authors report here. Therefore, the reader could expect the authors to take a stance on

uncertainty in time estimation, especially since the manuscript's abstract closes with this bold

statement: "Our study reveals brain networks that integrate multilevel temporal information,

offering insight into the hierarchical predictive coding of time." It is worth noting that the

manuscript's modeling work unrealistically implies that time is encoded *perfectly* in the brain, without any error....

The authors state in the Discussion (p 18, l. 18-20): "Therefore, although we believe that the

time estimation during waiting is unlikely to be equally precise throughout, we chose to use a

simpler model that does not include explicit blurring parameters or assumptions about

precision."

To the contrary, the non-blurred models that the authors identified as the best-fitting ones make a specific assumption about precision, i.e. that the brains studied in this experiment are perfect clocks that make no timing errors (and can perfectly estimate two hazard rates simultaneously). This seems to be a neurobiologically implausible assumption, given the literature on time estimation (Gibbon Psychol Rev 1977, Gallistel & Gibbon Psychol Rev 2000).

REVISION ROUND 2 - AUTHORS' RESPONSE:

Response 5: On using different blurring factors and parameters (also dddressing the third question below)

We sincerely appreciate the reviewer's comments regarding the role of uncertainty in time estimation. We acknowledge that our previous response may have unintentionally given the impression that temporal and probabilistic blurring were used solely to improve model fit. We would like to clarify that our intention in testing these different blurring factors was to evaluate competing hypotheses about the neural representation of uncertainty during time processing.

To our knowledge, the question of how the brain represents uncertainty in temporal prediction remains unresolved in the existing literature. For example, neural activity in the primate brain has been shown to closely correlate with temporally blurred HFs under both unimodal and bimodal distributions (Janssen & Shadlen, 2005). In long-foreperiod paradigms, where greater temporal uncertainty is expected, RTs have also been successfully explained by temporally blurred HFs (Bueti et al., 2010; Bueti & Macaluso, 2010). Conversely, other studies suggest that probabilistically blurred probability distribution itself (PDF) better account for human behavioral data (Grabenhorst et al., 2019, 2021). Still, unblurred HFs have also been effectively used to differentiate behavioral and neural responses across uniform and unimodal distributions (Herbst et al., 2018), in both explicit and implicit timing tasks (Coull & Nobre, 2008), and in modeling trial-by-trial neural responses to surprise in the foreperiod paradigm (Visalli et al., 2021).

These seemingly conflicting findings may reflect important differences across studies (Nobre & van Ede, 2018), including: (1) the statistical properties of the foreperiod distributions (e.g., mean and variance); (2) the temporal resolution of the experimental design; and (3) the use of catch trials. These factors can result in varied temporal prediction profiles, along with different types and magnitudes of uncertainty. For example, regarding the first point, in a bimodal distribution formed by overlapping two identical unimodal distributions, increasing the distance between the peaks can amplify temporal blurring. This is evident in the ~0.27- and ~1.93-second peaks in Janssen & Shadlen (2005) and the ~3.91- and ~13.75-second peaks in Bueti et al. (2010), compared to the 0.8- and 1.6-second peaks in the current study. For probabilistic blurring, the degree of blurring can be influenced by the standard deviation of the distribution, a narrower distribution produces sharper changes over time, whereas a flatter distribution leads to more gradual changes.

Regarding the second point, even when foreperiod distributions have identical means and standard deviations, the overall range can still affect uncertainty. For instance, a longer range (0- 20 seconds, as in Bueti et al., 2010) can introduce greater temporal uncertainty than a shorter range (0.4-2 seconds), as used in our study. Furthermore, it is also reasonable to infer that both temporal and probabilistic blurring may coexist and interact with each other when using probability distributions with different means, standard deviations, and time ranges. In a side note, temporal resolutionm, defined by the time step used in the analysis (e.g., 0.1-second vs. 0.01-second), can also influence the dynamics of the HF, particularly around the later peaks. This effect is illustrated in our simulations based on Bueti et al. (2010) and our own study, as well as in comparisons with Herbst et al. (2018) (see green arrows in the figures below).

--- FIG

Third, the inclusion of catch trials in some foreperiod paradigms not only introduces additional uncertainty regarding whether the target will occur but also complicates the definition of the probability distribution for target trials. Regarding the former, Grabenhorst et al. (2021) modeled RT by incorporating both the uncertainty of target occurrence and probabilistic uncertainty (i.e., probabilistic blurring), raising important questions about how these two forms of uncertainty interact in the brain. Regarding the latter, distribution estimation becomes more complex, as catch trials may either be omitted or interpreted as extremely long foreperiods within the distribution.

Returning to our study, which aimed to test whether the brain can learn and integrate both unconditional and conditional temporal statistics, we implemented distinct foreperiod distributions that inevitably introduced different types of uncertainty across conditions. We found that no single blurring factor consistently outperformed others across FP1 and FP2, further supporting our first and second points.

Importantly, we do not claim that the original HF consistently yielded the best-fitting model. Rather, we emphasize that the integration of multilevel temporal information was consistently observed in any blurring factors used. Notably, all better-fitting models were based on hazard functions—either temporally or probabilistically blurred. Consistent with prior neurophysiological findings demonstrating neural encoding of original hazard functions, our results revealed significant associations between RT/neural activity and original HF predictors. This suggests that the brain may perform HF-like computations as a mechanism for temporal prediction, with different forms of uncertainty modulating these estimates in a context-dependent and potentially parallel manner.

REVISION ROUND 3 - COMMENT REVIEWER #1:

The authors acknowledge that "the question of how the brain represents uncertainty in temporal prediction remains unresolved in the existing literature" and in the following sections conjecture different hypotheses about why this may be the case. So the authors identified the uncertainty in time estimation as an important aspect in temporal prediction. Yet despite the authors' lengthy musings that follow this insight, the manuscript at hand does not offer any tangible progress on this aspect.

Instead, the manuscript leaves the reader puzzled by two points:

1) Their temporally / probabilistically blurred HF fits fail to yield consistent results across conditions. They conclude that they are not appropriate or some other thing is needed, yet they do not offer a modeling solution to this problem. At this point the reader wonders: What is the major contribution of this manuscript: There is a novel task and new data but there is just a weak fit of the nonblurred Hazard Function and a non-conclusive investigation of different hypotheses about temporal uncertainty.

2) The blurring should account for noise in the data and, if the temporal blurring/probabilistic blurring hypotheses are on the right track, there should be an improved model fit. Yet the results are inconsistent across conditions. This further questions the basic modeling approach of this manuscript. The off-the-shelf Hazard Function appears to be a wrong model choice in the first place.

REVISION ROUND 2 - AUTHORS' RESPONSE:

Once again, we fully acknowledge the reviewer's important point: non-blurred models imply a simplified assumption that time is encoded without error. While this assumption allows us to isolate core computational principles, it underrepresents the role of uncertainty.

REVISION ROUND 3 - COMMENT REVIEWER #1:

The authors cannot claim to have "isolated core computational primitives" The fits are unconvincing both quantitatively (R-squqred, beta) and qualitatively (visual inspection).

REVISION ROUND 2 - AUTHORS' RESPONSE:

Its modulatory effects should be further investigated in future research, such as using foreperiod ranges of different lengths (to test temporal blurring) or manipulating the standard deviation of distributions (to test probabilistic blurring). Accordingly, we have explicitly discussed this limitation in the revised Discussion to better reflect the complexity of the underlying temporal processes.

Discussion section:

"Still, in long-foreperiod paradigms, where greater temporal uncertainty would be expected, RTs have been well explained by temporally blurred HFs (Bueti et al., 2010). These seemingly conflicting findings across studies may reflect key differences across studies (Nobre & van Ede, 2018), including: (1) the statistical properties of the foreperiod distributions (e.g., mean and standard deviation), (2) the temporal resolution of the experimental design, and (3) the use of catch trials. These factors can introduce varied temporal prediction profiles with different types and magnitudes of uncertainty. Regarding the first point, in a bimodal distribution created by overlapping two identical unimodal distributions, a greater distance between the peaks is expected to produce a stronger temporal blurring effect. For probabilistic blurring, the extent of the effect can be influenced by the distribution's standard deviation, for instance, a sharper distribution may lead to more rapid changes over time than a flatter one. For the second point, even when distributions share the same mean and standard deviation, the range of foreperiods can still affect uncertainty. For example, a longer range (e.g., 0-20 seconds, as in Bueti et al., 2010) introduces greater temporal uncertainty than a shorter range (e.g., 0.4-2 seconds, as used in our study). It is also reasonable to infer that temporal and probabilistic blurring may coexist and interact, potentially modulating one another, when using probability distributions with different means, standard deviations, and time ranges. Third, the inclusion of catch trials not only adds uncertainty regarding whether the target will occur but also complicates the definition of the underlying probability distribution. For instance, Grabenhorst et al. (2021) modeled RT by incorporating both the uncertainty of target occurrence and probabilistic uncertainty (i.e., probabilistic blurring), raising the question of how these two forms of uncertainty interact in the brain. In terms of probability distributions, catch trials may either be excluded or interpreted as extremely long foreperiods, making distribution estimation more complex.

This inconsistency further supports our earlier points: in our study, we implemented distinct foreperiod distributions to examine whether the brain can learn and integrate both unconditional and conditional temporal statistics. This design inevitably introduced different types of uncertainty across conditions. As shown in previous neurophysiological work, which demonstrates neural encoding of original hazard functions, and consistent with our findings, we suggest that the brain may rely on HF-like computations as a neural mechanism for temporal prediction. These computations may be modulated by different forms of uncertainty in a context-dependent and potentially parallel manner. Nevertheless, we emphasize that while using non-blurred models allows us to isolate core computational principles, under the simplifying assumption that time is encoded without error, it also underrepresents the role of uncertainty. The modulatory effects of uncertainty should be explored in future research, for example by varying foreperiod ranges (to examine temporal blurring) or manipulating the standard deviation of the distribution (to examine probabilistic blurring)." (Pages 18-20, Lines 18-14)

REVISION ROUND 2 - COMMENT REVIEWER #1:

In fact, it should worry the authors that all versions of their models (non-blurred, temporally

blurred, probabilistically blurred, Suppl. Fig. 8) yield rather small goddness-of-fit values (adj.

R2), as the authors acknowledge. The authors speculate that this may be due to the number of trials per participant which they say may be too small. Three questions:

1) If this is the case, then how can the main results of the manuscript be conclusive?

REVISION ROUND 2 - AUTHORS' RESPONSE:

Response 6: Trial Numbers

We appreciate the reviewer's concern. When designing our experiment, we determined the number of trials, 600 for FP1 and 600 for FP2, based primarily on prior studies. Specifically, we referred to: (1) Herbst et al. (2018), in which each participant completed a total of 546 trials during EEG recordings; and (2) Bueti et al. (2010), which included 600 trials in a training session and 200 trials in a testing session during fMRI acquisition. Notably, they reported no substantial differences between the results from the two sessions. Based on these references, we specified the rationale for our chosen trial numbers in the Methods section.

REVISION ROUND 3 - COMMENT REVIEWER #1:

The authors state that they presented 600 trials for FP1 and for FP2. When inspecting the data more closely (see my above comments to Fig. 2), it became obvious that there is not much of a patterned RT modulation over the longer half of foreperiods. Why may this be in light of the large trial numbers?

The experiment is structured in blocks 1 to 4 and in their manuscript, the authors write: "During the experiment, the four blocks were delivered in a random order and repeated three times, each with a different random order, resulting in a total of 12 block representations (run)."

If I understand this correctly, then there were no two same consecutive blocks in the experiment. Could then one explanation for this lack of a clear effect be that the authors presented too few trials per block before randomly switching to another block ("block" = experimental condition)? Compared to work in the literature, where several hundreds of trials are presented before switching condition, this would render the design underpowered: the authors may have switched condition before participants reached the end of learning.

REVISION ROUND 2 - AUTHORS' RESPONSE:

In addition, we acknowledge that our earlier attribution of low conditional R2 values to limited trial numbers per participant was speculative and not supported by direct evidence. To evaluate this more rigorously, we conducted a bootstrapping analysis to assess the stability of R2 values within our dataset. Specifically, for each participant and for both FP1 and FP2, we resampled their trials with replacement and re-estimated the R2 value 1,000 times to generate a bootstrap distribution and corresponding 95% confidence intervals. This allowed us to assess whether the observed R2 values were statistically stable and representative. Please see the results below and in Supplementary Figures 11 and 12. Our assessment is based on two criteria: (1) The bootstrap distribution of R2 should be unimodal. Bimodal or uniform distributions would suggest instability. While skewness toward zero may reflect limited informativeness, it does not necessarily imply instability. (2) The actual R2 value should fall within the 95% confidence interval of the bootstrap distribution and ideally be close to the bootstrap mean. Across participants, these criteria were generally met for both FP1 and FP2, suggesting that the number of trials per participant was sufficient to yield statistically stable R2 estimates. We believe these findings reinforce the robustness of our conclusions.

REVISION ROUND 3 - COMMENT REVIEWER #1:

The results of the boot-strapping analysis reinforces the robustness of the authors' conclusions only in the sense that the HF model really offers only a weak model fit.

REVISION ROUND 2 - AUTHORS' RESPONSE:

Methods section:

"Each participant completed a total of 600 FP1 trials and 600 FP2 trials, with the number of trials determined primarily based on prior studies (Bueti et al., 2010; Herbst et al., 2018). We also confirmed the stability of individual results using bootstrap resampling, which showed that the distribution of individual R-squared values was unimodal, with the actual values close to the bootstrap mean and within the 95% confidence intervals (Supplementary Figures 11 and 12)." (Page 28, Lines 6-11)

---Suppl Fig 11

---Suppl Fig 12

REVISION ROUND 2 - COMMENT REVIEWER #1

2) Do the HF models fit the group-level RT well (see Preface)?

3) Is the hazard function an adequate model for these data at all?

REVISION ROUND 2 - AUTHORS' RESPONSE:

These two questions have been addressed in Responses 1-5.

REVISION ROUND 2 - COMMENT REVIEWER #1:

Comment 6

The authors convincingly addressed this comment.

Comment 7

In the Discussion (p. 20 l. 2-3), the authors write: "Here, it is important to clarify again that the

integration of two probabilities was supported by the optimal model performance with the

inclusion of the interaction term (i.e., HFU * HFC)."

The term "optimal" is inappropriate here. The authors probably want to express that the model with the interaction term yielded the best fit relative to the other models that did not feature such a term. So this is a relative statement and should be formulated as such. The current sentence implies that the model is performing optimally, which does not make sense.

REVISION ROUND 2 - AUTHORS' RESPONSE:

We thank the reviewer for pointing this out. Accordingly, we have revised the sentence to: "Here, it is important to clarify again that the integration of two probabilities was supported by the model including the interaction term (i.e., HFU * HFC), which yielded the best fit relative to the other models that did not feature such a term." (Page 18, Line 5-12)

REVISION ROUND 3 - COMMENT REVIEWER #1:

Ok.

REVISION ROUND 2 - COMMENT REVIEWER #1:

Comment 8

The authors convincingly addressed this comment.

Comment 9

The authors convincingly addressed this comment.

Comment 10

The authors convincingly addressed this comment.

Reviewer #2: In my view the authors have addressed all concerns with the current revision. The requested plots (RT over FP) and additional analyses are sufficiently convincing that the model explains important computational aspects of temporal predictions as the authors claim, even though not all variability in RT is captured (as in many existing studies).

It is inherently difficult to design probabilistic variations of foreperiods, and regress these to RT, due to the multiple facets of temporal prediction, and contributions to RT such as autocorrelations over trials, lapses, ..

My recommendation is to accept the paper for publication. I belief it is an important addition to the literature.

---

## [Editor Report · Decision Letter 2]

9 Oct 2025

Dear Dr Huang,

Thank you for the submission of your revised Research Article "Human brain integrates both unconditional and conditional timing statistics to guide expectation and behavior" for publication in PLOS Biology. On behalf of my colleagues and the Academic Editor, Ruth de Diego Balaguer, I am pleased to say that we can in principle accept your manuscript for publication, provided you address any remaining formatting and reporting issues. These will be detailed in an email you should receive within 2-3 business days from our colleagues in the journal operations team; no action is required from you until then. Please note that we will not be able to formally accept your manuscript and schedule it for publication until you have completed any requested changes.

When you attend to those request to come, please also add a citation of the location of the source data clearly in all relevant main and supplementary Figure legends, e.g. “The data underlying this Figure can be found in https://doi.org/10.17605/OSF.IO/VEDHP”."

PRESS

Sincerely, 

Christian

Christian Schnell, PhD

Senior Editor

PLOS Biology

cschnell@plos.org